# Robust Finite-Time Stability for Uncertain Discrete-Time Stochastic Nonlinear Systems with Time-Varying Delay

**DOI:** 10.3390/e24060828

**Published:** 2022-06-14

**Authors:** Xikui Liu, Wencong Li, Jiqiu Wang, Yan Li

**Affiliations:** 1College of Mathematics and Systems Science, Shandong University of Science and Technology, Qingdao 266590, China; liuxikui@sdust.edu.cn (X.L.); lwccy2020@163.com (W.L.); wangjiqiu0828@163.com (J.W.); 2Department of Fundamental Courses, Shandong University of Science and Technology, Jinan 250031, China

**Keywords:** discrete-time stochastic system, finite-time stability, nonlinear perturbations, uncertain parameters, time-varying delay

## Abstract

The main concern of this paper is finite-time stability (FTS) for uncertain discrete-time stochastic nonlinear systems (DSNSs) with time-varying delay (TVD) and multiplicative noise. First, a Lyapunov–Krasovskii function (LKF) is constructed, using the forward difference, and less conservative stability criteria are obtained. By solving a series of linear matrix inequalities (LMIs), some sufficient conditions for FTS of the stochastic system are found. Moreover, FTS is presented for a stochastic nominal system. Lastly, the validity and improvement of the proposed methods are shown with two simulation examples.

## 1. Introduction

Time-delays are general in many actual systems, for instance, circuits, neural network systems, biological medicine, building structure and multi-agent systems [1,2,3,4]. However, a time-delay may reduce the performance of dynamic systems and even lead to system instability. Therefore, how to eliminate the adverse effects caused by time-delay on the system is an important consideration. A new Lyapunov method was proposed to study the stability of the system [5]. In [6], Zhang et al. introduced a reciprocally convex matrix inequality to analyze the stability of TVD systems. A FTS or stabilization criterion was given for linear time-delay systems (TDSs) through bounded linear time-varying feedback [7]. The practical stability of TVD-positive systems was analyzed by designing a controller [8]. Long et al. considered the stability of linear TDS via the quadratic function negative deterministic method [9].

In actual engineering, it is common for the system to be interfered with by some nonlinear factors. The control analysis of nonlinear TDSs becomes more important. In order to obtain a greater upper limit of TVD, a new less-conservative stability criterion for nonlinear perturbed TDS was proposed in [10]. The authors of [11] developed robust stability of switching systems with nonlinear disturbances and interval TVD. Finite-time control problems for nonlinear systems with TVD and external interference were discussed in [12]. In general, there is often uncertainty in the parameters of the system model. Kang et al. [13] studied the FTS of discrete-time nonlinear systems with interval TVD. On the basis of [13], Stojanovic proposed a less conservative FTS criterion for discrete-time nonlinear systems with TVD and uncertain terms [14].

The most basic concepts of system stability include FTS [15,16,17] and Lyapunov asymptotic stability (LAS) [18,19]. FTS and LAS are different in two ways. First, FTS studies the state behavior of a system within a limited time interval, and LAS studies the state behavior of a system in an infinite time interval. Secondly, the former research needs to give the limit of the system state in advance, while the latter does not need to give a fixed value in advance. In [20], Dorato introduced the concept of FTS. In a sense, if the state of the system under the initial condition does not exceed the specified range, then the system is called FTS within a certain time interval. In recent years, the research on FTS has attracted much attention. Thus far, many new results on FTS have been published. Shi et al. extended the discrete Jensen-based inequality by establishing a new weighted summation inequality and proposed new criteria for the FTS of nonlinear TDS [21]. The concepts of FTS and stabilization are extended to continuous additive TVD systems [22].

In the above-mentioned literature, the dynamic systems are definite; however, in the actual industrial production process, random phenomena are widespread [23]. From the perspective of theory or practical application, the control problem of stochastic systems is always a difficult issue. In [24], Yu et al. studied the FTS and $H_{\infty }$ control problem for stochastic nonlinear systems. Wang et al. discussed the FTS of stochastic nonlinear systems by constructing Hamiltonian functions [25]. The stochastic FTS of linear semi-Markov jump systems was developed by designing a class of state feedback controllers [26].

Yan et al. proposed a less conservative stability criterion through a model-dependent method and gave the FTS and stabilization conditions of Markov jump $It\hat{o}$ stochastic systems [27]. The authors in [28] considered the FTS problem of stochastic linear discrete-time TDS with multiplicative noise under state feedback control [29] discussed the finite-time guaranteed performance control of random mean-field systems and gave the criterion of FTS for the closed-loop system. However, there is no literature to discuss the FTS of uncertain DSNS with multiplicative noise and time-delay.

The results of FTS are extended to uncertain DSNS with TVD in this paper. The followings are the major contributions: First, a new augmented time-varying LKF is proposed, which contains a power function $\zeta^{k-j-1}$. A special finite sum inequality (NFSI) is used to process the forward difference of LKF with double summation terms. This NFSI is equivalent to Jensen’s inequality but reduces conservatism to a certain extent. Secondly, one-dimensional random processes, namely a sequence of one-dimensional independent white noise processes defined in probability space $\left(\Omega ,F,P,F_{n}\right)$, are considered as an external disturbance of the system. An algorithm is presented to reduce the influence of external interference on the stability of uncertain DSNS.

In addition, the uncertain parameters and nonlinear perturbations are transformed into linearity through the inequality calculation. The remainder is arranged as follows: some preparatory knowledge for the discrete-time stochastic system, the requisite lemmas and definitions are given in Section 2. The FTS of discrete time-varying stochastic uncertain system with nonlinear perturbations is studied, and then the FTS of a nominal system is considered in Section 3. By constructing an LKF with a power function and a new summation term, the criteria to guarantee the FTS of the discrete-time stochastic system are given. Simulation examples are presented to demonstrate the validity of the results in Section 4. Section 5 is the conclusion of this paper.

Notation: In this paper, $R^{n}$ means the *n*-dimensional Euclidean space, $R^{n\times m}$ is all real matrices with dimensions of n×m. $N_{0}=\left\{-h_{M},-h_{M}+1,\cdots ,-1,0\right\}$, N={1,2,3,⋯,N}. For matrix $P\in R^{n\times m}$, $\lambda_{max}\left(P\right)$ represents the maximum eigenvalue, and $\lambda_{min}\left(P\right)$ denotes the minimum eigenvalue of matrix *P*. P>0(P≥0) means *P* is positive definite (positive semi-definite) matrix. The ∗ sign in the matrix indicates the symmetry item.

## 2. Problem Statement and Preliminaries

Consider an uncertain DSNS with TVD as follows
(1)$$\begin{array}{c} \left\{\begin{array}{c} \begin{array}{cc} x(k+1)= & \left(M+\Delta M\left(k\right)\right)x\left(k\right)+\left(M_{d}+\Delta M_{d}\left(k\right)\right)x\left(k-h\left(k\right)\right)\\ & +e_{1}\left(x\left(k\right),k\right)+e_{2}\left(x\left(k-h\left(k\right)\right),k\right)+Dx\left(k\right)\omega \left(k\right),\\ x(l)= & \varphi \left(l\right),\;l\in N_{0}, \end{array} \end{array}\right. \end{array}$$
where $x\left(k\right)\in R^{n}$ is the state vector, ω(k) is a one-dimensional random variable defined on the probability space $\left(\Omega ,F,P,F_{n}\right)$. $M,\;M_{d},\;D\in R^{n\times n}$ are constant matrices with appropriate dimensions. It is supposed that E{ω(l)}=0, $E\left\{\omega \left(l\right)\omega \left(k\right)\right\}=\delta_{lk}$, when l≠k, $\delta_{lk}=0$; when l=k, $\delta_{lk}=1$, where E{·} represents mathematical expectation, and $\delta_{lk}$ is a Kronecker function. h(k) is the interval TVD and satisfies $0<h_{m}\leq h\left(k\right)\leq h_{M}$. φ(l) is a vector-valued initial sequence such that
$$\underset{l\in N_{0}}{\sup}{\left(\varphi \left(l+1\right)-\varphi \left(l\right)\right)}^{T}\left(\varphi \left(l+1\right)-\varphi \left(l\right)\right)\leq \delta .$$

$e_{1}\left(x\left(k\right),k\right)$ and $e_{2}\left(x\left(k-h\left(k\right)\right),k\right)$ are nonlinear disturbances, and *H* and $H_{d}$ are known constant matrices satisfying
$$\begin{array}{cc} {e_{1}}^{T}\left(x\left(k\right),k\right)e_{1}\left(x\left(k\right),k\right) & \leq x^{T}\left(k\right)H^{T}Hx\left(k\right),\\ {e_{2}}^{T}\left(x\left(k-h\left(k\right)\right),k\right)e_{2}\left(x\left(k-h\left(k\right)\right),k\right) & \leq x^{T}\left(k-h\left(k\right)\right){H_{d}}^{T}H_{d}x\left(k-h\left(k\right)\right). \end{array}$$

ΔM(k), $\Delta M_{d}\left(k\right)$ and Δ(k) are unknown time-varying matrices that are uniformly bounded in the norm. *F*, $F_{d}$ and *G* are known matrices with
$$\begin{array}{c} \left[\begin{array}{cc} \Delta M(k) & \Delta M_{d}\left(k\right) \end{array}\right]=G\Delta \left(k\right)\left[\begin{array}{cc} F & F_{d} \end{array}\right],\;\Delta^{T}\left(k\right)\Delta \left(k\right)\leq I. \end{array}$$

Let z(k) represent an unknown variable, and the system (Equation 1) can be written as follows:$$\begin{array}{c} \left\{\begin{array}{c} \begin{array}{cc} x(k+1)= & Mx\left(k\right)+M_{d}x\left(k-h\left(k\right)\right)+Gz\left(k\right)+e_{1}\left(x\left(k\right),k\right)\\ & +e_{2}\left(x\left(k-h\left(k\right)\right),k\right)+Dx\left(k\right)\omega \left(k\right),\\ z(k)= & \Delta \left(k\right)(Fx\left(k\right)+F_{d}x\left(k-h\left(k\right)\right)),\\ x(l)= & \varphi \left(l\right),\;l\in N_{0}. \end{array} \end{array}\right. \end{array}$$

For system (Equation 1), if the perturbations and uncertainties are not considered, it can be expressed as:(2)$$\begin{array}{c} \left\{\begin{array}{c} \begin{array}{cc} x(k+1) & =Mx\left(k\right)+M_{d}x\left(k-h\left(k\right)\right)+Dx\left(k\right)\omega \left(k\right),\\ x(l) & =\varphi \left(l\right),\;l\in N_{0}, \end{array} \end{array}\right. \end{array}$$
which is called a nominal system.

In order to analyze the FTS of system (Equation 1), the following definition and lemmas are introduced.

**Definition** **1.**
*For given scalars 0≤α≤β, system (Equation 1) is FTS subject to (α,β,N), n∈N, if*

$$\underset{l\in N_{0}}{\sup}{∥\varphi \left(l\right)∥}^{2}\leq \alpha \Rightarrow {E\{∥x\left(k\right)∥}^{2}\}<\beta .$$



**Remark** **1.**
*It is worth noting that α and β are all used to describe the state variables remain on the given limits. β is influenced by α as a parameter adjustment in the simulation; however, that does not mean that β only depends on α. In addition, different from the plain stability, the characteristics of finite-time stability are as follows: First, the initial condition is confined to a prescribed limit. Second, the state trajectory does not exceed the specified value over a finite-time interval instead of an infinite-time interval.*


**Remark** **2.**
*Compared with the continuous-time system [30,31,32], the FTS of the discrete-time stochastic systems [33,34,35] has received much less attention because the analysis can be simplified to the robust stability of linear systems with uncertainties but no delay. That is to say, in the discrete-time case, the presence of time-delay can be modeled by extending the system state with past variables, i.e., $X\left(k\right)={\left[x\left(k\right),x\left(k-1\right),\cdot \cdot \cdot ,x\left(k-hM\right)\right]}^{T}$. However, the optimization algorithms are easier to complete on the computer as well as the growing applications in certain engineering fields; therefore, the stability analysis of discrete-time stochastic systems is meaningful. In the next sections, we extend the analysis of FTS to stochastic systems.*


**Lemma** **1**([36])**.**
*For any matrices with appropriate dimensions L>0, $L^{T}>0$, $L\in R^{m\times m}$, $S\in R^{n\times m}$, scalar ζ>0 and $h_{1},h_{2}\in N$, $h_{2}>h_{1}$, the following inequality holds*
(3)$$-\sum_{j=k-h_{2}}^{k-h_{1}-1}\zeta^{k-j}y^{T}\left(j\right)Ly\left(j\right)\leq \xi^{T}\left(k\right)\varepsilon SL^{-1}S^{T}\xi \left(k\right)+2\xi^{T}\left(k\right)S\left(x\left(k-h_{1}\right)-x\left(k-h_{2}\right)\right),$$*where y(j)=x(j+1)−x(j), $\xi \left(k\right)\in R^{n\times 1}$ is the state vector and ε>0 is a constant defined as follows*
$$\begin{array}{c} \begin{array}{c} \varepsilon =\left\{\begin{array}{cc} h_{2}-h_{1}, & \zeta =1,\\ \left(\zeta^{-h_{1}}-\zeta^{-h_{2}}\right)/\left(\zeta -1\right), & \zeta \neq 1. \end{array}\right. \end{array} \end{array}$$

**Lemma** **2**([37])**.**
*(Schur complement) For the given symmetric matrix $T=\left[\begin{array}{cc} T_{11} & T_{12}\\ {}* & T_{22} \end{array}\right]$, where $T_{11}\in R^{n\times n}$, the following three conditions are equivalent:*
$$\begin{array}{cc} \; & (1)\;T<0;\\ \; & \left(2\right)\;T_{11}<0,T_{22}-T_{12}^{T}T_{11}^{-1}T_{12}<0;\\ \; & \left(3\right)\;T_{22}<0,T_{11}-T_{12}T_{22}^{-1}T_{12}^{T}<0. \end{array}$$

## 3. Main Results

In this section, the FTS and RAS problems for system (Equation 1) are discussed, and the FTS criteria for the nominal system (Equation 2) are presented.

### 3.1. FTS for Stochastic Systems with Nonlinear Disturbances and Uncertain Parameters

In this section, stochastic system (Equation 1) is considered with uncertain parameters and nonlinear disturbances, the sufficient conditions of FTS for system (Equation 1) are given first.

**Theorem** **1.**
*Given constant ζ>1 and positive constants $\epsilon ,\;\epsilon_{d},\;\mu$, $\lambda_{1},\lambda_{2},\cdots ,\lambda_{8}$, then the uncertain DSNS (Equation 1) with TVD is FTS subject to {α,β,N} if there exist matrices $A={\left[A_{1}^{T}\;A_{2}^{T}\;\cdots \;A_{7}^{T}\right]}^{T},$$B={\left[B_{1}^{T}\;B_{2}^{T}\;\cdots \;B_{7}^{T}\right]}^{T}$ and $C={\left[C_{1}^{T}\;C_{2}^{T}\;\cdots \;C_{7}^{T}\right]}^{T}$ as well as positive definite matrices $P,W_{1},W_{2},$$Q_{1},$$Q_{2}\in R^{n\times n}$, such that all of the following conditions hold:*

(4)
$$\left[\begin{array}{cccc} {\left[\Theta_{ij}\right]}_{i,j=1,2,\cdots ,7} & \varepsilon_{1}A & \varepsilon_{1}B & \varepsilon_{2}C\\ {}* & -\varepsilon_{1}Q_{1} & 0 & 0\\ {}* & * & -\varepsilon_{1}Q_{1} & 0\\ {}* & * & * & -\varepsilon_{2}Q_{2} \end{array}\right]<0,$$


(5)
$$\begin{array}{cc} \lambda_{1}I<P<\lambda_{2}I,\;\lambda_{3}I & <W_{1}<\lambda_{4}I,\;\lambda_{5}I<W_{2}<\lambda_{6}I,\\ Q_{1}< & \lambda_{7}I,\;Q_{2}<\lambda_{8}I, \end{array}$$


(6)
$$\begin{array}{c} \zeta^{N}\left[\alpha \left(\lambda_{2}+\eta_{1}\lambda_{4}+\eta_{2}\lambda_{6}\right)+\delta \left(\eta_{3}\lambda_{7}+\eta_{4}\lambda_{8}\right)\right]-\beta \left[\lambda_{1}+\eta_{1}\lambda_{3}+\eta_{2}\lambda_{5}\right]<0, \end{array}$$

*where*

$$\begin{array}{cc} \Theta_{11}= & \;M^{T}PM-\zeta P+W_{1}+W_{2}+{\left(M-I\right)}^{T}Q_{12}\left(M-I\right)+\epsilon H^{T}H+\mu F^{T}F\\ \; & +C_{1}+C_{1}^{T}+D^{T}PD+D^{T}Q_{12}D,\\ \Theta_{12}= & \;M^{T}PM_{d}+{\left(M-I\right)}^{T}Q_{12}M_{d}+\mu F^{T}F_{d}+A_{1}-B_{1}+C_{2}^{T},\\ \Theta_{13}= & \;B_{1}-C_{1}+C_{3}^{T},\;\Theta_{14}=-A_{1}+C_{4}^{T},\\ \Theta_{15}= & \;M^{T}PG+{\left(M-I\right)}^{T}Q_{12}G+C_{5}^{T},\;\Theta_{16}=M^{T}P+{\left(M-I\right)}^{T}Q_{12}+C_{6}^{T},\\ \Theta_{17}= & \;M^{T}P+{\left(M-I\right)}^{T}Q_{12}+C_{7}^{T},\\ \Theta_{22}= & \;M_{d}^{T}PM_{d}+M_{d}^{T}Q_{12}M_{d}+\epsilon_{d}H_{d}^{T}H_{d}+\mu F_{d}^{T}F_{d}+A_{2}+A_{2}^{T}-B_{2}-B_{2}^{T},\\ \Theta_{23}= & \;A_{3}^{T}+B_{2}-B_{3}^{T}-C_{2},\;\Theta_{24}=-A_{2}+A_{4}^{T}-B_{4}^{T},\\ \Theta_{25}= & \;M_{d}^{T}PG+M_{d}^{T}Q_{12}G+A_{5}^{T}-B_{5}^{T},\;\Theta_{26}=M_{d}^{T}P+M_{d}^{T}Q_{12}+A_{6}^{T}-B_{6}^{T},\\ \Theta_{27}= & \;M_{d}^{T}P+M_{d}^{T}Q_{12}+A_{7}^{T}-B_{7}^{T},\;\Theta_{33}=-\zeta^{h_{m}}W_{2}+B_{3}+B_{3}^{T}-C_{3}-C_{3}^{T},\\ \Theta_{34}= & \;-A_{3}+B_{4}^{T}-C_{4}^{T},\;\Theta_{35}=B_{5}^{T}-C_{5}^{T},\;\Theta_{36}=B_{6}^{T}-C_{6}^{T},\\ \Theta_{37}= & \;B_{7}^{T}-C_{7}^{T},\;\Theta_{44}=-\zeta^{h_{M}}W_{1}-A_{4}-A_{4}^{T},\;\Theta_{45}=-A_{5}^{T},\\ \Theta_{46}= & \;-A_{6}^{T},\;\Theta_{47}=-A_{7}^{T},\;\Theta_{55}=G^{T}PG+G^{T}Q_{12}G-\mu I,\\ \Theta_{56}= & \;G^{T}P+G^{T}Q_{12},\;\Theta_{57}=G^{T}P+G^{T}Q_{12},\;\Theta_{66}=P+Q_{12}-\epsilon I,\\ \Theta_{67}= & \;P+Q_{12},\;\Theta_{77}=P+Q_{12}-\epsilon_{d}I,\;Q_{12}=\left(h_{M}-h_{m}\right)Q_{1}+h_{m}Q_{2}, \end{array}$$


(7)
$$\begin{array}{c} \begin{array}{c} \varepsilon_{1}=\left\{\begin{array}{cc} h_{M}-h_{m}, & \zeta =1,\\ \left(\zeta^{-h_{m}}-\zeta^{-h_{M}}\right)/\left(\zeta -1\right), & \zeta \neq 1, \end{array}\right.\;\varepsilon_{2}=\left\{\begin{array}{cc} h_{m}, & \zeta =1,\\ \left(1-\zeta^{-h_{m}}\right)/\left(\zeta -1\right), & \zeta \neq 1. \end{array}\right. \end{array} \end{array}$$


(8)
$$\begin{array}{c} \begin{array}{cc} \eta_{1}= & \;\left\{\begin{array}{cc} h_{M}, & \zeta =1,\\ \left(\zeta^{h_{M}}-1\right)/\left(\zeta -1\right), & \zeta \neq 1, \end{array}\right.\;\eta_{2}=\left\{\begin{array}{cc} h_{m}, & \zeta =1,\\ \left(\zeta^{h_{m}}-1\right)/\left(\zeta -1\right), & \zeta \neq 1, \end{array}\right.\\ \eta_{3}= & \;\left\{\begin{array}{cc} \frac{h_{M}\left(h_{M}+1\right)}{2}-\frac{h_{m}\left(h_{m}+1\right)}{2}, & \zeta =1,\\ \left(\zeta^{h_{M}+1}-\zeta^{h_{m}+1}-\left(\zeta -1\right)\left(h_{M}-h_{m}\right)\right)/{\left(\zeta -1\right)}^{2}, & \zeta \neq 1, \end{array}\right.\\ \eta_{4}= & \;\left\{\begin{array}{cc} \frac{h_{m}\left(h_{m}+1\right)}{2}, & \zeta =1,\\ \left(\zeta \left(\zeta^{h_{m}}-1\right)-\left(\zeta -1\right)h_{m}\right)/{\left(\zeta -1\right)}^{2}, & \zeta \neq 1. \end{array}\right. \end{array} \end{array}$$



**Proof.** Consider the following LKF (k∈N),
(9)$$V\left(k\right)=\sum_{s=1}^{3}V_{s}\left(k\right),$$
where

$$\begin{array}{cc} V_{1}\left(k\right) & =x^{T}\left(k\right)Px\left(k\right),\\ V_{2}\left(k\right) & =\sum_{j=k-h_{M}}^{k-1}\zeta^{k-j-1}x^{T}\left(j\right)W_{1}x\left(j\right)+\sum_{j=k-h_{m}}^{k-1}\zeta^{k-j-1}x^{T}\left(j\right)W_{2}x\left(j\right),\\ V_{3}\left(k\right) & =\sum_{i=-h_{M}}^{-h_{m}-1}\sum_{j=k+i}^{k-1}\zeta^{k-j-1}y^{T}\left(j\right)Q_{1}y\left(j\right)+\sum_{i=-h_{m}}^{-1}\sum_{j=k+i}^{k-1}\zeta^{k-j-1}y^{T}\left(j\right)Q_{2}y\left(j\right). \end{array}$$

Along the trajectory of system (Equation 1), the forward differences of $V_{1}\left(k\right)$, $V_{2}\left(k\right)$ and $V_{3}\left(k\right)$ are obtained as follows
(10)E{ΔV1(k)}=EV1(k+1)−V1(k)=ExT(k+1)Px(k+1)+(ζ−1)V1(k)−ζV1(k)=ExT(k)(MTPM+DTPD−ζP)x(k)+(ζ−1)V1(k)+2xT(k)MTPMdx(k−h(k))+2xT(k)MTPGz(k)+2xT(k)MTPe1(x(k),k)+2xT(k)MTPe2(x(k−h(k)),k)+xT(k−h(k))MdTPMdx(k−h(k))+2xT(k−h(k))MdTPGz(k)+2xT(k−h(k))MdTPe1(x(k),k)+zT(k)GTPGz(k)+2xT(k−h(k))MdTPe2(x(k−h(k)),k)+2zT(k)GTPe1(x(k),k)+2zT(k)GTPe2(x(k−h(k)),k)+e1T(x(k),k)Pe1(x(k),k)+2e1T(x(k),k)Pe2(x(k−h(k)),k)=+e2T(x(k−h(k)),k)Pe2(x(k−h(k)),k),
(11)$$\begin{array}{cc} \; & \;E\left\{\Delta V_{2}\left(k\right)\right\}=\;E\left\{V_{2}\left(k+1\right)-V_{2}\left(k\right)\right\}\\ =\; & E\{\sum_{j=k-h_{M}+1}^{k}\zeta^{k-j}x^{T}\left(j\right)W_{1}x\left(j\right)+\sum_{j=k-h_{m}+1}^{k}\zeta^{k-j}x^{T}\left(j\right)W_{2}x\left(j\right)\\ \; & -\sum_{j=k-h_{M}}^{k-1}\zeta^{k-j-1}x^{T}\left(j\right)W_{1}x\left(j\right)-\sum_{j=k-h_{m}}^{k-1}\zeta^{k-j-1}x^{T}\left(j\right)W_{2}x\left(j\right)\}\\ =\; & E\{\zeta \sum_{j=k-h_{M}}^{k-1}\zeta^{k-j-1}x^{T}\left(j\right)W_{1}x\left(j\right)+x^{T}\left(k\right)W_{1}x\left(k\right)\\ \; & -\zeta^{h_{M}}x^{T}\left(k-h_{M}\right)W_{1}x\left(k-h_{M}\right)\\ \; & +\zeta \sum_{j=k-h_{m}}^{k-1}\zeta^{k-j-1}x^{T}\left(j\right)W_{2}x\left(j\right)+x^{T}\left(k\right)W_{2}x\left(k\right)\\ \; & -\zeta^{h_{m}}x^{T}\left(k-h_{m}\right)W_{2}x\left(k-h_{m}\right)-V_{2}\left(k\right)\}\\ =\; & E\left\{\left(\zeta -1\right)V_{2}\left(k\right)+x^{T}\left(k\right)\left(W_{1}+W_{2}\right)x\left(k\right)\right.\\ \; & -\zeta^{h_{M}}x^{T}\left(k-h_{M}\right)W_{1}x\left(k-h_{M}\right)\left.-\zeta^{h_{m}}x^{T}\left(k-h_{m}\right)W_{2}x\left(k-h_{m}\right)\right\}, \end{array}$$
(12)$$\begin{array}{cc} \; & \;E\left\{\Delta V_{3}\left(k\right)\right\}=\;E\left\{V_{3}\left(k+1\right)-V_{3}\left(k\right)\right\}\\ =\; & E\{\sum_{i=-h_{M}}^{-h_{m}-1}\sum_{j=k+i+1}^{k}\zeta^{k-j}y^{T}\left(j\right)Q_{1}y\left(j\right)+\sum_{i=-h_{m}}^{-1}\sum_{j=k+i+1}^{k}\zeta^{k-j}y^{T}\left(j\right)Q_{2}y\left(j\right)\\ \; & -\sum_{i=-h_{M}}^{-h_{m}-1}\sum_{j=k+i}^{k-1}\zeta^{k-j-1}y^{T}\left(j\right)Q_{1}y\left(j\right)-\sum_{i=-h_{m}}^{-1}\sum_{j=k+i}^{k-1}\zeta^{k-j-1}y^{T}\left(j\right)Q_{2}y\left(j\right)\}\\ =\; & E\{\sum_{i=-h_{M}}^{-h_{m}-1}\sum_{j=k+i}^{k-1}\zeta^{k-j}y^{T}\left(j\right)Q_{1}y\left(j\right)+\sum_{i=-h_{M}}^{-h_{m}-1}y^{T}\left(k\right)Q_{1}y\left(k\right)\\ \; & -\sum_{i=-h_{M}}^{-h_{m}-1}\zeta^{-i}y^{T}\left(k+i\right)Q_{1}y\left(k+i\right)+\sum_{i=-h_{m}}^{-1}\sum_{j=k+i}^{k-1}\zeta^{k-j}y^{T}\left(j\right)Q_{2}y\left(j\right)\\ \; & +\sum_{i=-h_{m}}^{-1}y^{T}\left(k\right)Q_{2}y\left(k\right)-\sum_{i=-h_{m}}^{-1}\zeta^{-i}y^{T}\left(k+i\right)Q_{2}y\left(k+i\right)-V_{3}\left(k\right)\}, \end{array}$$
then
(13)$$\begin{array}{cc} E\{\Delta V_{3}\left(k\right)\}=\; & E\{\left(h_{M}-h_{m}\right)y^{T}\left(k\right)Q_{1}y\left(k\right)-\sum_{j=k-h_{M}}^{k-h_{m}-1}\zeta^{k-j}y^{T}\left(j\right)Q_{1}y\left(j\right)\\ \; & +h_{m}y^{T}\left(k\right)Q_{2}y\left(k\right)-\sum_{j=k-h_{m}}^{k-1}\zeta^{k-j}y^{T}\left(j\right)Q_{2}y\left(j\right)+\left(\zeta -1\right)V_{3}\left(k\right)\}\\ =\; & E\{\left(\zeta -1\right)V_{3}\left(k\right)+y^{T}\left(k\right)\left[\left(h_{M}-h_{m}\right)Q_{1}+h_{m}Q_{2}\right]y\left(k\right)\\ \; & -\sum_{j=k-h_{M}}^{k-h_{m}-1}\zeta^{k-j}y^{T}\left(j\right)Q_{1}y\left(j\right)-\sum_{j=k-h_{m}}^{k-1}\zeta^{k-j}y^{T}\left(j\right)Q_{2}y\left(j\right)\}. \end{array}$$From Lemma 1 and (Equation 13), we have
(14)E−∑j=k−hMk−hm−1ζk−jyT(j)Q1y(j)=E−∑j=k−hMk−h(k)−1ζk−jyT(j)Q1y(j)−∑j=k−h(k)k−hm−1ζk−jyT(j)Q1y(j)≤EξT(k)ε1′AQ1−1ATξ(k)+2ξT(k)A(x(k−h(k))−x(k−hM))=+ξT(k)ε1″BQ1−1BTξ(k)+2ξT(k)B(x(k−hm)−x(k−h(k)))≤EξT(k)ε1AQ1−1ATξ(k)+2ξT(k)A0I0−I000ξ(k)=+ξT(k)ε1BQ1−1BTξ(k)+2ξT(k)B0−II0000ξ(k)=EξT(k)(Π1+ε1AQ1−1AT+ε1BQ1−1BT)ξ(k),
(15)$$\begin{array}{cc} \; & E\left\{-\sum_{j=k-h_{m}}^{k-1}\zeta^{k-j}y^{T}\left(j\right)Q_{2}y\left(j\right)\right\}\\ \leq \; & E\left\{\xi^{T}\left(k\right)\varepsilon_{2}CQ_{2}^{-1}C^{T}\xi \left(k\right)+2\xi^{T}\left(k\right)C\left(x\left(k\right)-x\left(k-h_{m}\right)\right)\right\}\\ =\; & E\left\{\xi^{T}\left(k\right)\varepsilon_{2}CQ_{2}^{-1}C^{T}\xi \left(k\right)+2\xi^{T}\left(k\right)C\left[\begin{array}{ccccccc} I & 0 & -I & 0 & 0 & 0 & 0 \end{array}\right]\xi \left(k\right)\right\}\\ =\; & E\left\{\xi^{T}\left(k\right)\left(\Pi_{2}+\varepsilon_{2}CQ_{2}^{-1}C^{T}\right)\xi \left(k\right)\right\}, \end{array}$$
where
$$\begin{array}{c} \xi \left(n\right)=[x^{T}\left(k\right)\;x^{T}\left(k-h\left(k\right)\right)\;x^{T}\left(k-h_{m}\right)\;x^{T}\left(k-h_{M}\right)\\ z^{T}\left(k\right)\;{e_{1}}^{T}\left(x\left(k\right),k\right)\;{e_{2}}^{T}{\left(x\left(k-h\left(k\right)\right),k\right)]}^{T}, \end{array}$$
$$\begin{array}{cc} \Pi_{1}=\; & \left[\begin{array}{ccccccc} 0 & \Pi_{12} & B_{1} & -A_{1} & 0 & 0 & 0\\ {}* & \Pi_{22} & \Pi_{23} & \Pi_{24} & A_{5}^{T}-B_{5}^{T} & A_{6}^{T}-B_{6}^{T} & A_{7}^{T}-B_{7}^{T}\\ {}* & * & B_{3}+B_{3}^{T} & -A_{3}+B_{4}^{T} & B_{5}^{T} & B_{6}^{T} & B_{7}^{T}\\ {}* & * & * & -A_{4}-A_{4}^{T} & -A_{5}^{T} & -A_{6}^{T} & -A_{7}^{T}\\ {}* & * & * & * & 0 & 0 & 0\\ {}* & * & * & * & * & 0 & 0\\ {}* & * & * & * & * & * & 0 \end{array}\right], \end{array}$$
$$\begin{array}{cc} \Pi_{12} & =A_{1}-B_{1},\;\Pi_{22}=A_{2}-B_{2}+A_{2}^{T}-B_{2}^{T},\\ \Pi_{23} & =B_{2}+A_{3}^{T}-B_{3}^{T},\;\Pi_{24}=-A_{2}+A_{4}^{T}-B_{4}^{T}, \end{array}$$
$$\begin{array}{cc} \Pi_{2}=\; & \left[\begin{array}{ccccccc} C_{1}+C_{1}^{T} & C_{2}^{T} & -C_{1}+C_{3}^{T} & C_{4}^{T} & C_{5}^{T} & C_{6}^{T} & C_{7}^{T}\\ {}* & 0 & -C_{2} & 0 & 0 & 0 & 0\\ {}* & * & -C_{3}-C_{3}^{T} & -C_{4}^{T} & -C_{5}^{T} & -C_{6}^{T} & -C_{7}^{T}\\ {}* & * & * & 0 & 0 & 0 & 0\\ {}* & * & * & * & 0 & 0 & 0\\ {}* & * & * & * & * & 0 & 0\\ {}* & * & * & * & * & * & 0 \end{array}\right],\\ \varepsilon_{1}^{{}^{\prime }}=\; & \left(\zeta^{-h(k)}-\zeta^{-h_{M}}\right)/\left(\zeta -1\right)\leq \left(\zeta^{-h_{m}}-\zeta^{-h_{M}}\right)/\left(\zeta -1\right)=\varepsilon_{1},\\ \varepsilon_{1}^{{}^{\prime \prime }}=\; & \left(\zeta^{-h_{m}}-\zeta^{-h(k)}\right)/\left(\zeta -1\right)\leq \left(\zeta^{-h_{m}}-\zeta^{-h_{M}}\right)/\left(\zeta -1\right)=\varepsilon_{1}. \end{array}$$From Equations (Equation 10)–(Equation 15) and condition (Equation 16)
(16)$$\begin{array}{c} \begin{array}{cc} \epsilon x^{T}\left(k\right)H^{T}Hx\left(k\right)- & \epsilon {e_{1}}^{T}\left(x\left(k\right),k\right)e_{1}\left(x\left(k\right),k\right)\geq 0,\\ \epsilon_{d}x^{T}\left(k-h\left(k\right)\right)H_{d}^{T}H_{d}x\left(k-h\left(k\right)\right) & -\epsilon_{d}{e_{2}}^{T}\left(x\left(k-h\left(k\right)\right),k\right)e_{2}\left(x\left(k-h\left(k\right)\right),k\right)\geq 0, \end{array} \end{array}$$
we deduce that
$$\begin{array}{cc} \; & \;\;E\left\{\Delta V(k)\right\}\\ \leq & E\left\{\left(\zeta -1\right)V\left(k\right)+x^{T}\left(k\right)[M^{T}PM+D^{T}PD-\zeta P\right.\\ \; & +W_{1}+W_{2}+{\left(M-I\right)}^{T}Q_{12}\left(M-I\right)+D^{T}Q_{12}D]x\left(k\right)\\ \; & +2x^{T}\left(k\right)\left(M^{T}PM_{d}+{\left(M-I\right)}^{T}Q_{12}M_{d}\right)x\left(k-h\left(k\right)\right)\\ \; & +2x^{T}\left(k\right)\left(M^{T}PG+{\left(M-I\right)}^{T}Q_{12}G\right)z\left(k\right)\\ \; & +2x^{T}\left(k\right)\left(M^{T}P+{\left(M-I\right)}^{T}Q_{12}\right)e_{1}\left(x\left(k\right),k\right)\\ \; & +2x^{T}\left(k\right)\left(M^{T}P+{\left(M-I\right)}^{T}Q_{12}\right)e_{2}\left(x\left(k-h\left(k\right)\right),k\right)\\ \; & +x^{T}\left(k-h\left(k\right)\right)\left(M_{d}^{T}PM_{d}+M_{d}^{T}Q_{12}M_{d}\right)x\left(k-h\left(k\right)\right)\\ \; & +2x^{T}\left(k-h\left(k\right)\right)\left(M_{d}^{T}PG+M_{d}^{T}Q_{12}G\right)z\left(k\right)\\ \; & +2x^{T}\left(k-h\left(k\right)\right)\left(M_{d}^{T}P+M_{d}^{T}Q_{12}\right)e_{1}\left(x\left(k\right),k\right)\\ \; & +2x^{T}\left(k-h\left(k\right)\right)\left(M_{d}^{T}P+M_{d}^{T}Q_{12}\right)e_{2}\left(x\left(k-h\left(k\right)\right),k\right)\\ \; & +z^{T}\left(k\right)\left(G^{T}PG+G^{T}Q_{12}G\right)z\left(k\right)\\ \; & +2z^{T}\left(k\right)\left(G^{T}P+G^{T}Q_{12}\right)e_{1}\left(x\left(k\right),k\right)\\ \; & +2z^{T}\left(k\right)\left(G^{T}P+G^{T}Q_{12}\right)e_{2}\left(x\left(k-h\left(k\right)\right),k\right)\\ \; & +{e_{1}}^{T}\left(x\left(k\right),k\right)\left(P+Q_{12}\right)e_{1}\left(x\left(k\right),k\right)\\ \; & +2{e_{1}}^{T}\left(x\left(k\right),k\right)\left(P+Q_{12}\right)e_{2}\left(x\left(k-h\left(k\right)\right),k\right)\\ \; & +{e_{2}}^{T}\left(x\left(k-h\left(k\right)\right),k\right)\left(P+Q_{12}\right)e_{2}\left(x\left(k-h\left(k\right)\right),k\right)\\ \; & -\zeta^{h_{M}}x^{T}\left(k-h_{M}\right)W_{1}x\left(k-h_{M}\right)\\ \; & -\zeta^{h_{m}}x^{T}\left(k-h_{m}\right)W_{2}x\left(k-h_{m}\right)\\ \; & +\xi^{T}\left(k\right)(\Pi_{1}+\Pi_{2}+\varepsilon_{1}AQ_{1}^{-1}A^{T}\\ \; & +\varepsilon_{1}BQ_{1}^{-1}B^{T}+\varepsilon_{2}CQ_{2}^{-1}C^{T})\xi \left(k\right)\\ \; & +\epsilon x^{T}\left(k\right)H^{T}Hx\left(k\right)-\epsilon {e_{1}}^{T}\left(x\left(k\right),k\right)e_{1}\left(x\left(k\right),k\right)\\ \; & +\epsilon_{d}x^{T}\left(k-h\left(k\right)\right)H_{d}^{T}H_{d}x\left(k-h\left(k\right)\right)\\ \; & -\epsilon_{d}{e_{2}}^{T}\left(x\left(k-h\left(k\right)\right),k\right)e_{2}\left(x\left(k-h\left(k\right)\right),k\right)\\ \; & +\mu x^{T}\left(k\right)F^{T}Fx\left(k\right)-2\mu x^{T}\left(k\right)F^{T}F_{d}x\left(k-h\left(k\right)\right)\\ \; & \left.+\mu x^{T}\left(k-h\left(k\right)\right)F_{d}^{T}F_{d}x\left(k-h\left(k\right)\right)-\mu z^{T}\left(k\right)z\left(k\right)\right\}, \end{array}$$
and it is rewritten that
EΔV(k)≤E(ζ−1)V(k)+ξT(k)(([Θij]i,j=1,2,⋯,7)+ε1AQ1−1AT=+ε1BQ1−1BT+ε2CQ2−1CT)ξ(k).From Schur’s complement, (Equation 4) is equivalent to inequality (Equation 17)
(17)$$\begin{array}{c} \left({\left[\Theta_{ij}\right]}_{i,j=1,2,\cdots ,7}\right)+\varepsilon_{1}AQ_{1}^{-1}A^{T}+\varepsilon_{1}BQ_{1}^{-1}B^{T}+\varepsilon_{2}CQ_{2}^{-1}C^{T}<0. \end{array}$$Then,
$$\begin{array}{c} E\left\{\Delta V(k)-(\zeta -1)V(k)\right\}<0, \end{array}$$
by calculating, one further finds
(18)$$\begin{array}{c} E\left\{V(k)\right\}<\zeta^{k}E\left\{V(0)\right\},k\in N. \end{array}$$According to the definition of E{V(k)},
(19)EV(0)=ExT(0)Px(0)+∑j=−hM−1ζ−j−1xT(j)W1x(j)+∑j=−hm−1ζ−j−1xT(j)W2x(j)=+∑i=−hM−hm−1∑j=i−1ζ−j−1yT(j)Q1y(j)+∑i=−hm−1∑j=i−1ζ−j−1yT(j)Q2y(j)≤Eαλmax(P)+αλmax(W1)∑j=−hM−1ζ−j−1+αλmax(W2)∑j=−hm−1ζ−j−1=+δλmax(Q1)∑i=−hM−hm−1∑j=i−1ζ−j−1+δλmax(Q2)∑i=−hm−1∑j=i−1ζ−j−1=α(λmax(P)+η1λmax(W1)+η2λmax(W2))+δ(η3λmax(Q1)+η4λmax(Q2)),
(20)EV(k)>Eλmin(P)∥x(k)∥2+λmin(W1)∑j=k−hMk−1ζk−j−1∥x(j)∥2=+λmin(W2)∑j=k−hmk−1ζk−j−1∥x(j)∥2.In addition,
(21)$$\begin{array}{cc} \; & \zeta^{N}\left[\alpha \left(\lambda_{max}\left(P\right)+\eta_{1}\lambda_{max}\left(W_{1}\right)+\eta_{2}\lambda_{max}\left(W_{2}\right)\right)+\delta \left(\eta_{3}\lambda_{max}\left(Q_{1}\right)+\eta_{4}\lambda_{max}\left(Q_{2}\right)\right)\right]\\ <\; & \beta [\lambda_{min}\left(P\right)+\eta_{1}\lambda_{min}\left(W_{1}\right)+\eta_{2}\lambda_{min}\left(W_{2}\right)]. \end{array}$$It follows from (Equation 21) that
(22)$$\begin{array}{cc} \; & E\left\{\lambda_{min}{\left(P\right)∥x\left(k\right)∥}^{2}+\lambda_{min}\left(W_{1}\right)\sum_{j=k-h_{M}}^{k-1}\zeta^{k-j-1}∥x\left(j\right){∥}^{2}+\lambda_{min}\left(W_{2}\right)\sum_{j=k-h_{m}}^{k-1}\zeta^{k-j-1}{∥x\left(j\right)∥}^{2}\right\}\\ <\; & E\left\{V(k)\right\}<\zeta^{N}\left[\alpha \left(\lambda_{max}\left(P\right)+\eta_{1}\lambda_{max}\left(W_{1}\right)+\eta_{2}\lambda_{max}\left(W_{2}\right)\right)+\delta \left(\eta_{3}\lambda_{max}\left(Q_{1}\right)+\eta_{4}\lambda_{max}\left(Q_{2}\right)\right)\right]\\ <\; & \beta [\lambda_{min}\left(P\right)+\eta_{1}\lambda_{min}\left(W_{1}\right)+\eta_{2}\lambda_{min}\left(W_{2}\right)]. \end{array}$$From (Equation 5), (Equation 6) and (Equation 22), $E\{∥x\left(k\right){∥}^{2}\}<\beta$, k∈N. Thus, the stochastic system (Equation 1) is FTS from Definition 1. The proof is completed. □

**Remark** **3.**
*For stochastic system (Equation 1), the conservativeness of the FTS criterion is generally restricted by inequalities (Equation 18)–(Equation 20). We consider the LKF with the power function $\zeta^{k-j}$ and find E{V(k)}<ζE{V(k−1)}, then $E\left\{V\left(k\right)\right\}<\zeta^{k}E\left\{V\left(0\right)\right\}$, $E\left\{V\left(0\right)\right\}<\Gamma_{1}$ and $E\left\{V\left(k\right)\right\}>\Gamma_{2}$, where $\Gamma_{1}$ and $\Gamma_{2}$ correspond to the estimates of the upper and lower bounds of E{V(0)} and E{V(k)}, respectively. The estimates are related to α, β, N, δ, $h_{m}$, $h_{M}$ and ζ.*


**Remark** **4.**
*The FTS criterion considered in this paper is less conservative than literature [13], which dealt with summation terms $\sum_{i=-h_{M}}^{-1}\sum_{j=k+i}^{k-1}\zeta^{k-j-1}y^{T}\left(j\right)Q_{1}y\left(j\right)$ and $\sum_{i=-h_{M}}^{-h_{m}-1}\sum_{j=k+i}^{k-1}\zeta^{k-j-1}y^{T}\left(j\right)Q_{2}y\left(j\right)$ by Jensen inequality. In this paper, a new finite sum inequality with time-delay states (Lemma 1) is used to term $\sum_{i=-h_{M}}^{-1}\sum_{j=k+i}^{k-1}\zeta^{k-j-1}y^{T}\left(j\right)Q_{1}y\left(j\right)$, instead of using the discrete Jensen inequality [38] or the Wirtinger-based inequality [39]. Inequality (Equation 3) is introduced to deal with term $\sum_{i=-h_{M}}^{-h_{m}-1}\sum_{j=k+i}^{k-1}\zeta^{k-j-1}y^{T}\left(j\right)Q_{2}y\left(j\right)$ rather than the free weighting matrix method presented in [40]. Considering the influence of nonlinear factors, a new finite sum inequality is used to deal with the forward difference of Lyapunov functional so that the finite sums $-E\left\{\sum_{j=k-h_{M}}^{k-h_{m}-1}\zeta^{k-j}y^{T}\left(j\right)Q_{1}y\left(j\right)\right\}$ and $-E\left\{\sum_{j=k-h_{m}}^{k-1}\zeta^{k-j}y^{T}\left(j\right)Q_{2}y\left(j\right)\right\}$ are estimated accurately.*


### 3.2. FTS for Nominal Systems

In this section, the criterion of the nominal system (Equation 2) FTS is given. In particular, we consider the FTS of the nominal system when h(k)=h.

**Corollary** **1.**
*Given constant ζ>1 and the positive constants $\lambda_{i}\left(i=1,2,\cdots ,8\right)$, the nominal system (Equation 2) is FTS subject to {α,β,N} if there exist matrices $A={\left[A_{1}^{T}\;A_{2}^{T}\;A_{3}^{T}\;A_{4}^{T}\right]}^{T},$$B={\left[B_{1}^{T}\;B_{2}^{T}\;B_{3}^{T}\;B_{4}^{T}\right]}^{T}$ and $C={\left[C_{1}^{T}\;C_{2}^{T}\;C_{3}^{T}\;C_{4}^{T}\right]}^{T}$ as well as positive definite matrices $P,W_{1},W_{2},Q_{1},$$Q_{2}\in R^{n\times n}$, such that the following conditions hold:*

(23)
$$\left[\begin{array}{cccc} {\left[{\bar{\Theta }}_{ij}\right]}_{i,j=1,\cdots ,4} & \varepsilon_{1}A & \varepsilon_{1}B & \varepsilon_{2}C\\ {}* & -\varepsilon_{1}Q_{1} & 0 & 0\\ {}* & * & -\varepsilon_{1}Q_{1} & 0\\ {}* & * & * & -\varepsilon_{2}Q_{2} \end{array}\right]<0,$$


λ1I<P<λ2I,λ3I<W1<λ4I,λ5I<W2<λ6I,Q1<λ7I,Q2<λ8I,


ζN[α(λ2+η1λ4+η2λ6)+δ(η3λ7+η4λ8)]−β[λ1+η1λ3+η2λ5]<0,

*where*

$$\begin{array}{cc} {\bar{\Theta }}_{11}= & \;M^{T}PM-\zeta P+W_{1}+W_{2}+{\left(M-I\right)}^{T}Q_{12}\left(M-I\right)\\ \; & +C_{1}+C_{1}^{T}+D^{T}PD+D^{T}Q_{12}D,\\ {\bar{\Theta }}_{12}= & \;M^{T}PM_{d}+{\left(M-I\right)}^{T}Q_{12}M_{d}+A_{1}-B_{1}+C_{2}^{T},\\ {\bar{\Theta }}_{13}= & \;B_{1}-C_{1}+C_{3}^{T},\;{\bar{\Theta }}_{14}=-A_{1}+C_{4}^{T},\\ {\bar{\Theta }}_{22}= & \;M_{d}^{T}PM_{d}+M_{d}^{T}Q_{12}M_{d}+A_{2}+A_{2}^{T}-B_{2}-B_{2}^{T},\\ {\bar{\Theta }}_{23}= & \;A_{3}^{T}+B_{2}-B_{3}^{T}-C_{2},\;{\bar{\Theta }}_{24}=-A_{2}+A_{4}^{T}-B_{4}^{T},\\ {\bar{\Theta }}_{33}= & \;-\zeta^{h_{m}}W_{2}+B_{3}+B_{3}^{T}-C_{3}-C_{3}^{T},\;{\bar{\Theta }}_{34}=-A_{3}+B_{4}^{T}-C_{4}^{T},\\ {\bar{\Theta }}_{44}= & \;-\zeta^{h_{M}}W_{1}-A_{4}-A_{4}^{T}, \end{array}$$

*constants $\varepsilon_{1}$, $\varepsilon_{2}$, $\eta_{1}$, $\eta_{2}$, $\eta_{3}$ and $\eta_{4}$ satisfy (Equation 7) and (Equation 8).*


**Proof.** Let us select the LKF (Equation 9). Then, along the trajectory of the nominal system (Equation 2), the forward difference of E{ΔV(k)} is obtained as follows
(24)$$\begin{array}{cc} E\left\{\Delta V(k)\right\}\leq \; & E\left\{\left(\zeta -1\right)V\left(k\right)+x^{T}\left(k\right)[M^{T}PM+D^{T}PD+D^{T}Q_{12}D-\zeta P\right.\\ \; & +{\left(M-I\right)}^{T}Q_{12}\left(M-I\right)+W_{1}+W_{2}]x\left(k\right)\\ \; & +2x^{T}\left(k\right)\left(M^{T}PM_{d}+{\left(M-I\right)}^{T}Q_{12}M_{d}\right)x\left(k-h\left(k\right)\right)\\ \; & +x^{T}\left(k-h\left(k\right)\right)\left(M_{d}^{T}PM_{d}+M_{d}^{T}Q_{12}M_{d}\right)x\left(k-h\left(k\right)\right)\\ \; & -\sum_{j=k-h_{M}}^{k-h_{m}-1}\zeta^{k-j}y^{T}\left(j\right)Q_{1}y\left(j\right)-\sum_{j=k-h_{m}}^{k-1}\zeta^{k-j}y^{T}\left(j\right)Q_{2}y\left(j\right)\\ \; & -\zeta^{h_{M}}x^{T}\left(k-h_{M}\right)W_{1}x\left(k-h_{M}\right)\left.-\zeta^{h_{m}}x^{T}\left(k-h_{m}\right)W_{2}x\left(k-h_{m}\right)\right\}. \end{array}$$From Lemma 1, we find
(25)$$\begin{array}{c} -\sum_{j=k-h_{M}}^{k-h_{m}-1}\zeta^{k-j}y^{T}\left(j\right)Q_{1}y\left(j\right)\leq {\bar{\xi }}^{T}\left({\bar{\Pi }}_{1}+\varepsilon_{1}AQ_{1}^{-1}A^{T}+\varepsilon_{1}BQ_{1}^{-1}B^{T}\right)\bar{\xi }, \end{array}$$
(26)$$\begin{array}{c} -\sum_{j=k-h_{m}}^{k-1}\zeta^{k-j}y^{T}\left(j\right)Q_{2}y\left(j\right)\leq {\bar{\xi }}^{T}\left({\bar{\Pi }}_{2}+\varepsilon_{2}CQ_{2}^{-1}C^{T}\right)\bar{\xi }, \end{array}$$
where $\bar{\xi }={\left[\begin{array}{cccc} x^{T}\left(k\right) & x^{T}\left(k-h\left(k\right)\right) & x^{T}\left(k-h_{m}\right) & x^{T}\left(k-h_{M}\right) \end{array}\right]}^{T}$,
$$\begin{array}{cc} {\bar{\Pi }}_{1}=\; & \left[\begin{array}{cccc} 0 & A_{1}-B_{1} & B_{1} & -A_{1}\\ {}* & A_{2}+A_{2}^{T}-B_{2}-B_{2}^{T} & B_{2}+A_{3}^{T}-B_{3}^{T} & -A_{2}+A_{4}^{T}-B_{4}^{T}\\ {}* & * & B_{3}+B_{3}^{T} & -A_{3}+B_{4}^{T}\\ {}* & * & * & -A_{4}-A_{4}^{T} \end{array}\right], \end{array}$$
$$\begin{array}{cc} {\bar{\Pi }}_{2}=\; & \left[\begin{array}{cccc} C_{1}+C_{1}^{T} & C_{2}^{T} & -C_{1}+C_{3}^{T} & C_{4}^{T}\\ {}* & 0 & -C_{2} & 0\\ {}* & * & -C_{3}-C_{3}^{T} & -C_{4}^{T}\\ {}* & * & * & 0 \end{array}\right]. \end{array}$$Combining inequalities (Equation 24)–(Equation 26), it is inferred that
EΔV(k)≤E(ζ−1)V(k)+ξ¯T(k)(([Θ¯ij]i,j=1,⋯,4)+ε1AQ1−1AT=+ε1BQ1−1BT+ε2CQ2−1CT)ξ¯(k).By using the Schur complement and condition (Equation 23), inequality (Equation 27) holds
(27)$$\begin{array}{c} \left({\left[{\bar{\Theta }}_{ij}\right]}_{i,j=1,\cdots ,4}\right)+\varepsilon_{1}AQ_{1}^{-1}A^{T}+\varepsilon_{1}BQ_{1}^{-1}B^{T}+\varepsilon_{2}CQ_{2}^{-1}C^{T}<0, \end{array}$$
and then $E\left\{V\left(k\right)\right\}<\zeta^{k}E\left\{V\left(0\right)\right\},\;k\in N$. Furthermore, the proof of the latter part is similar to Theorem 1 and is here omitted. □

We take the constant time-delay (h(k)=h) as a special case, and the following corollary is obtained.

**Corollary** **2.**
*Given constant ζ>1 and positive constants $\lambda_{i}\left(i=1,2,\cdots ,5\right)$, then the nominal system (Equation 2) with h(k)=h is FTS subject to {α,β,N} if there exist matrices $A={\left[A_{1}^{T}\;A_{2}^{T}\right]}^{T}$ as well as positive definite symmetric matrices $P,W,Q\in R^{n\times n}$, such that*

$$\left[\begin{array}{cc} \underline{\Theta } & \varepsilon A\\ & -\varepsilon Q \end{array}\right]<0,$$


λ1I<P<λ2I,λ3I<W<λ4I,Q<λ5I,


ζN[α(λ2+η1λ4)+δη2λ5]−β(λ1+η1λ3)<0,

*where*

$$\begin{array}{cc} {\underline{\Theta }}_{11}= & \;M^{T}PM-\zeta P+W+h{\left(M-I\right)}^{T}Q\left(M-I\right)\\ \; & +A_{1}+A_{1}^{T}+D^{T}PD+D^{T}QD,\\ {\underline{\Theta }}_{12}= & \;M^{T}PM_{d}+h{\left(M-I\right)}^{T}QM_{d}-A_{1}+A_{2}^{T},\\ {\underline{\Theta }}_{22}= & \;M_{d}^{T}PM_{d}-\zeta^{h}W+hM_{d}^{T}QM_{d}-A_{2}-A_{2}^{T},\\ \underline{\xi }\left(k\right)= & \;{\left[\begin{array}{cc} x^{T}\left(k\right) & x^{T}\left(k-h\right) \end{array}\right]}^{T}, \end{array}$$


(28)
$$\begin{array}{c} \begin{array}{cc} \varepsilon = & \;\left\{\begin{array}{cc} h, & \zeta =1,\\ \left(1-\zeta^{-h}\right)/\left(\zeta -1\right), & \zeta \neq 1, \end{array}\right.\;\eta_{1}=\left\{\begin{array}{cc} h, & \zeta =1,\\ \left(\zeta^{h}-1\right)/\left(\zeta -1\right), & \zeta \neq 1, \end{array}\right.\\ \eta_{2}= & \;\left\{\begin{array}{cc} h(h+1)/2, & \zeta =1,\\ \left(\zeta \left(\zeta^{h}-1\right)-\left(\zeta -1\right)h\right)/{\left(\zeta -1\right)}^{2}, & \zeta \neq 1. \end{array}\right. \end{array} \end{array}$$



### 3.3. FTS for Stochastic Systems with Nonlinear Disturbances

In this section, the sufficient conditions for FTS and RAS of stochastic nonlinear systems (Equation 1) ($\Delta M\left(k\right)=\Delta M_{d}\left(k\right)=0$) are presented.

**Theorem** **2.**
*Given constant ζ>1 and positive constants $\epsilon ,\;\epsilon_{d}$, $\lambda_{i}\left(i=1,2,\cdots ,8\right)$, then the stochastic nonlinear system (Equation 1) with $\Delta M\left(k\right)=\Delta M_{d}\left(k\right)=0$ is FTS subject to {α,β,N} if there exist matrices $A={\left[A_{1}^{T}\;A_{2}^{T}\;\cdots \;A_{6}^{T}\right]}^{T}$, $B={\left[B_{1}^{T}\;B_{2}^{T}\;\cdots \;B_{6}^{T}\right]}^{T}$ and $C={\left[C_{1}^{T}\;C_{2}^{T}\;\cdots \;C_{6}^{T}\right]}^{T}$ as well as positive definite matrices $P,W_{1},W_{2},Q_{1},Q_{2}\in R^{n\times n}$ satisfying*

$$\left[\begin{array}{cccc} {\left[{\hat{\Theta }}_{ij}\right]}_{i,j=1,2,\cdots ,6} & \varepsilon_{1}A & \varepsilon_{1}B & \varepsilon_{2}C\\ {}* & -\varepsilon_{1}Q_{1} & 0 & 0\\ {}* & * & -\varepsilon_{1}Q_{1} & 0\\ {}* & * & * & -\varepsilon_{2}Q_{2} \end{array}\right]<0,$$


λ1I<P<λ2I,λ3I<W1<λ4I,λ5I<W2<λ6I,Q1<λ7I,Q2<λ8I,


ζN[α(λ2+η1λ4+η2λ6)+δ(η3λ7+η4λ8)]−β[λ1+η1λ3+η2λ5]<0,

*where*

$$\begin{array}{cc} {\hat{\Theta }}_{11}= & \;M^{T}PM-\zeta P+W_{1}+W_{2}+{\left(M-I\right)}^{T}Q_{12}\left(M-I\right)+\epsilon H^{T}H\\ \; & +C_{1}+C_{1}^{T}+D^{T}PD+D^{T}Q_{12}D,\\ {\hat{\Theta }}_{12}= & \;M^{T}PM_{d}+{\left(M-I\right)}^{T}Q_{12}M_{d}+A_{1}-B_{1}+C_{2}^{T},\\ {\hat{\Theta }}_{13}= & \;B_{1}-C_{1}+C_{3}^{T},\;{\hat{\Theta }}_{14}=-A_{1}+C_{4}^{T},\\ {\hat{\Theta }}_{15}= & \;M^{T}P+{\left(M-I\right)}^{T}Q_{12}+C_{5}^{T},\;{\hat{\Theta }}_{16}=M^{T}P+{\left(M-I\right)}^{T}Q_{12}+C_{6}^{T},\\ {\hat{\Theta }}_{22}= & \;M_{d}^{T}PM_{d}+M_{d}^{T}Q_{12}M_{d}+\epsilon_{d}H_{d}^{T}H_{d}+A_{2}+A_{2}^{T}-B_{2}-B_{2}^{T},\\ {\hat{\Theta }}_{23}= & \;A_{3}^{T}+B_{2}-B_{3}^{T}-C_{2},\;{\hat{\Theta }}_{24}=-A_{2}+A_{4}^{T}-B_{4}^{T},\\ {\hat{\Theta }}_{25}= & \;M_{d}^{T}P+M_{d}^{T}Q_{12}+A_{5}^{T}-B_{5}^{T},\;{\hat{\Theta }}_{26}=M_{d}^{T}P+M_{d}^{T}Q_{12}+A_{6}^{T}-B_{6}^{T},\\ {\hat{\Theta }}_{33}= & \;-\zeta^{h_{m}}W_{2}+B_{3}+B_{3}^{T}-C_{3}-C_{3}^{T},\;{\hat{\Theta }}_{34}=-A_{3}+B_{4}^{T}-C_{4}^{T},\\ {\hat{\Theta }}_{35}= & \;B_{5}^{T}-C_{5}^{T},\;{\hat{\Theta }}_{36}=B_{6}^{T}-C_{6}^{T},\;{\hat{\Theta }}_{44}=-\zeta^{h_{M}}W_{1}-A_{4}-A_{4}^{T},\\ {\hat{\Theta }}_{45}= & \;-A_{5}^{T},\;{\hat{\Theta }}_{46}=-A_{6}^{T},\;{\hat{\Theta }}_{55}=P+Q_{12}-\epsilon I,\\ {\hat{\Theta }}_{56}= & \;P+Q_{12},\;{\hat{\Theta }}_{66}=P+Q_{12}-\epsilon_{d}I,\\ \hat{\xi }\left(k\right)= & \;[x^{T}\left(k\right)\;x^{T}\left(k-h\left(k\right)\right)\;x^{T}\left(k-h_{m}\right)\;x^{T}\left(k-h_{M}\right)\\ & \;{e_{1}}^{T}\left(x\left(k\right),k\right)\;{e_{2}}^{T}{\left(x\left(k-h\left(k\right)\right),k\right)]}^{T}. \end{array}$$



When ζ=1 in Theorem 2, the RAS condition of the stochastic system (Equation 1) is obtained in the following corollary.

**Corollary** **3.**
*For the given constants ϵ and $\;\epsilon_{d}$, then the discrete time-varying stochastic nonlinear system (Equation 1) with $\Delta M\left(k\right)=\Delta M_{d}\left(k\right)=0$ is RAS if there exist matrices $A={\left[A_{1}^{T}\;A_{2}^{T}\;\cdots \;A_{6}^{T}\right]}^{T}$, $B={\left[B_{1}^{T}\;B_{2}^{T}\;\cdots \;B_{6}^{T}\right]}^{T}$ and $C={\left[C_{1}^{T}\;C_{2}^{T}\;\cdots \;C_{6}^{T}\right]}^{T}$, as well as positive definite symmetric matrices $P,W_{1},W_{2},Q_{1},$ and $Q_{2}\in R^{n\times n}$, such that the following inequalities hold:*

$$\left[\begin{array}{cccc} {\left[{\overset{ˇ}{\Theta }}_{ij}\right]}_{i,j=1,2,\cdots ,6} & (h_{M}-h_{m})A & (h_{M}-h_{m})B & h_{m}C\\ {}* & -\left(h_{M}-h_{m}\right)Q_{1} & 0 & 0\\ {}* & * & -\left(h_{M}-h_{m}\right)Q_{1} & 0\\ {}* & * & * & -h_{m}Q_{2} \end{array}\right]<0,$$

*where*

$$\begin{array}{cc} {\overset{ˇ}{\Theta }}_{11}= & \;M^{T}PM-P+W_{1}+W_{2}+{\left(M-I\right)}^{T}Q_{12}\left(M-I\right)+\epsilon H^{T}H\\ \; & +C_{1}+C_{1}^{T}+D^{T}PD+D^{T}Q_{12}D,\\ {\overset{ˇ}{\Theta }}_{12}= & \;M^{T}PM_{d}+{\left(M-I\right)}^{T}Q_{12}M_{d}+A_{1}-B_{1}+C_{2}^{T},\;{\overset{ˇ}{\Theta }}_{13}=B_{1}-C_{1}+C_{3}^{T},\\ {\overset{ˇ}{\Theta }}_{14}= & \;-A_{1}+C_{4}^{T},\;{\overset{ˇ}{\Theta }}_{15}=M^{T}P+{\left(M-I\right)}^{T}Q_{12}+C_{5}^{T},\\ {\overset{ˇ}{\Theta }}_{16}= & \;M^{T}P+{\left(M-I\right)}^{T}Q_{12}+C_{6}^{T},\\ {\overset{ˇ}{\Theta }}_{22}= & \;M_{d}^{T}PM_{d}+M_{d}^{T}Q_{12}M_{d}+\epsilon_{d}H_{d}^{T}H_{d}+A_{2}+A_{2}^{T}-B_{2}-B_{2}^{T},\\ {\overset{ˇ}{\Theta }}_{23}= & \;A_{3}^{T}+B_{2}-B_{3}^{T}-C_{2},\;{\overset{ˇ}{\Theta }}_{24}=-A_{2}+A_{4}^{T}-B_{4}^{T},\\ {\overset{ˇ}{\Theta }}_{25}= & \;M_{d}^{T}P+M_{d}^{T}Q_{12}+A_{5}^{T}-B_{5}^{T},\;{\overset{ˇ}{\Theta }}_{26}=M_{d}^{T}P+M_{d}^{T}Q_{12}+A_{6}^{T}-B_{6}^{T},\\ {\overset{ˇ}{\Theta }}_{33}= & \;-W_{2}+B_{3}+B_{3}^{T}-C_{3}-C_{3}^{T},\;{\overset{ˇ}{\Theta }}_{34}=-A_{3}+B_{4}^{T}-C_{4}^{T},\\ {\overset{ˇ}{\Theta }}_{35}= & \;B_{5}^{T}-C_{5}^{T},\;{\overset{ˇ}{\Theta }}_{36}=B_{6}^{T}-C_{6}^{T},\;{\overset{ˇ}{\Theta }}_{44}=-W_{1}-A_{4}-A_{4}^{T},\\ {\overset{ˇ}{\Theta }}_{45}= & \;-A_{5}^{T},\;{\overset{ˇ}{\Theta }}_{46}=-A_{6}^{T},\;{\overset{ˇ}{\Theta }}_{55}=P+Q_{12}-\epsilon I,\\ {\overset{ˇ}{\Theta }}_{56}= & \;P+Q_{12},\;{\overset{ˇ}{\Theta }}_{66}=P+Q_{12}-\epsilon_{d}I. \end{array}$$



In particular, when h(k)=h, the following corollary is drawn.

**Corollary** **4.**
*Given constant ζ>1 and positive constants $\epsilon ,\;\epsilon_{d}$, $\lambda_{i}\left(i=1,2,\cdots ,5\right)$, then the discrete stochastic nonlinear system (Equation 1) with ΔM(k)=0, $\Delta M_{d}\left(k\right)=0$ and h(k)=h is FTS subject to {α,β,N} if there exist matrix $A={\left[A_{1}^{T}\;A_{2}^{T}\;A_{3}^{T}\;A_{4}^{T}\right]}^{T}$ as well as positive definite matrices $P,W,Q\in R^{n\times n}$, such that all of the following conditions hold:*

(29)
$$\left[\begin{array}{cc} {\left[{\overset{˘}{\Theta }}_{ij}\right]}_{i,j=1,\cdots ,4} & \varepsilon A\\ {}* & -\varepsilon Q \end{array}\right]<0,$$


(30)
$$\lambda_{1}I<P<\lambda_{2}I,\;\lambda_{3}I<W<\lambda_{4}I,\;Q<\lambda_{5}I,$$


(31)
$$\zeta^{N}\left[\alpha \left(\lambda_{2}+\eta_{1}\lambda_{4}\right)+\delta \eta_{2}\lambda_{5}\right]-\beta \left(\lambda_{1}+\eta_{1}\lambda_{3}\right)<0,$$

*where*

$$\begin{array}{cc} {\overset{˘}{\Theta }}_{11}= & \;M^{T}PM-\zeta P+W+h{\left(M-I\right)}^{T}Q\left(M-I\right)+A_{1}+A_{1}^{T}\\ \; & +\epsilon H^{T}H+D^{T}PD+hD^{T}QD,\\ {\overset{˘}{\Theta }}_{12}= & \;M^{T}PM_{d}+h{\left(M-I\right)}^{T}QM_{d}-A_{1}+A_{2}^{T},\\ {\overset{˘}{\Theta }}_{13}= & \;M^{T}P+h{\left(M-I\right)}^{T}Q+A_{3}^{T},\\ {\overset{˘}{\Theta }}_{14}= & \;M^{T}P+h{\left(M-I\right)}^{T}Q+A_{4}^{T},\\ {\overset{˘}{\Theta }}_{22}= & \;M_{d}^{T}PM_{d}-\zeta^{h}W+hM_{d}^{T}QM_{d}-A_{2}-A_{2}^{T}+\epsilon_{d}H_{d}^{T}H_{d},\\ {\overset{˘}{\Theta }}_{23}= & \;M_{d}^{T}P+hM_{d}^{T}Q-A_{3}^{T},\;{\overset{˘}{\Theta }}_{24}=M_{d}^{T}P+hM_{d}^{T}Q-A_{4}^{T},\\ {\overset{˘}{\Theta }}_{33}= & \;P+hQ-\epsilon I,\;{\overset{˘}{\Theta }}_{34}=P+hQ,\;{\overset{˘}{\Theta }}_{44}=P+hQ-\epsilon_{d}I, \end{array}$$

*and the constants ε, $\eta_{1}$ and $\eta_{2}$ are defined by (Equation 28).*


**Proof.** First, choose the following LKF:
(32)$$V\left(k\right)=\sum_{s=1}^{3}V_{s}\left(k\right),$$
where
$$\begin{array}{cc} V_{1}\left(k\right) & =x^{T}\left(k\right)Px\left(k\right),\\ V_{2}\left(k\right) & =\sum_{j=k-h}^{k-1}\zeta^{k-j-1}x^{T}\left(j\right)Wx\left(j\right),\\ V_{3}\left(k\right) & =\sum_{i=-h}^{-1}\sum_{j=k+i}^{k-1}\zeta^{k-j-1}y^{T}\left(j\right)Qy\left(j\right). \end{array}$$Along the trajectory of system (Equation 1) with $\Delta M\left(k\right)=\Delta M_{d}\left(k\right)=0$, the forward difference of E{ΔV(k)} is obtained
(33)EΔV(k)≤E(ζ−1)V(k)+xT(k)[MTPM+DTPD+hDTQD−ζP+h(M−I)TQ(M−I)+W]x(k)+2xT(k)(MTPMd+h(M−I)QMd)x(k−h)+2xT(k)(MTP+h(M−I)Q)e1(x(k),k)+2xT(k)(MTP+h(M−I)Q)e2(x(k−h),k)+xT(k−h)(MdTPMd+hMdTQMd)x(k−h)+2xT(k−h)(MdTP+hMdTQ)e1(x(k),k)+2xT(k−h)(MdTP+hMdTQ)e2(x(k−h),k)+e1T(x(k),k)(P+hQ)e1(x(k),k)+2e1T(x(k),k)(P+hQ)e2(x(k−h),k)+e2T(x(k−h),k)(P+hQ)e2(x(k−h),k)=−∑j=k−hk−1ζk−jyT(j)Qy(j)−ζhxT(k−h)Wx(k−h).From Lemma 1, we find
(34)$$\begin{array}{c} -\sum_{j=k-h}^{k-1}\zeta^{k-j}y^{T}\left(j\right)Qy\left(j\right)\leq {\overset{˘}{\xi }}^{T}\left(\overset{˘}{\Pi }+\varepsilon AQ^{-1}A^{T}\right)\overset{˘}{\xi }, \end{array}$$
where $\overset{˘}{\xi }={\left[\begin{array}{cccc} x^{T}\left(k\right) & x^{T}\left(k-h\right) & {e_{1}}^{T}\left(x\left(k\right),k\right) & {e_{2}}^{T}\left(x\left(k-h\right),k\right) \end{array}\right]}^{T}$,
$$\begin{array}{cc} \overset{˘}{\Pi }=\; & \left[\begin{array}{ccccc} A_{1}+A_{1}^{T} & -A_{1}+A_{2}^{T} & A_{3}^{T} & A_{4}^{T}\\ {}* & -A_{2}-A_{2}^{T} & -A_{3}^{T} & -A_{4}^{T} & \\ {}* & * & 0 & 0\\ {}* & * & * & 0 \end{array}\right]. \end{array}$$By inequalities (Equation 33) and (Equation 34), we have
$$\epsilon x^{T}\left(k\right)H^{T}Hx\left(k\right)-\epsilon {e_{1}}^{T}\left(x\left(k\right),k\right)e_{1}\left(x\left(k\right),k\right)\geq 0,$$
$$\epsilon_{d}x^{T}\left(k-h\right)H_{d}^{T}H_{d}x\left(k-h\right)-\epsilon_{d}{e_{2}}^{T}\left(x\left(k-h\right),k\right)e_{2}\left(x\left(k-h\right),k\right)\geq 0.$$We conclude that
$$\begin{array}{cc} E\left\{\Delta V(k)\right\}\leq \; & E\left\{\left(\zeta -1\right)V\left(k\right)+{\overset{˘}{\xi }}^{T}\left(k\right)\left(\left({\left[{\overset{˘}{\Theta }}_{ij}\right]}_{i,j=1,\cdots ,4}\right)+\varepsilon AQ_{1}^{-1}A^{T}\right)\overset{˘}{\xi }\left(k\right)\right\}. \end{array}$$By the Schur complement and (Equation 29), (Equation 35) is found as
(35)$$\begin{array}{c} \left({\left[{\overset{˘}{\Theta }}_{ij}\right]}_{i,j=1,\cdots ,4}\right)+\varepsilon AQ_{1}^{-1}A^{T}<0. \end{array}$$Then, $E\left\{V\left(k\right)\right\}<\zeta^{k}E\left\{V\left(0\right)\right\},\;k\in N$.Furthermore, according to the definition of E{V(k)}, we can find
$$\begin{array}{cc} E\left\{V(0)\right\}=\; & E\left\{x^{T}\left(0\right)Px\left(0\right)+\sum_{j=-h}^{-1}\zeta^{-j-1}x^{T}\left(j\right)Wx\left(j\right)\right.\left.+\sum_{i=-h}^{-1}\sum_{j=i}^{-1}\zeta^{-j-1}y^{T}\left(j\right)Qy\left(j\right)\right\}\\ \leq \; & E\left\{\alpha \lambda_{max}\left(P\right)+\alpha \lambda_{max}\left(W\right)\sum_{j=-h}^{-1}\zeta^{-j-1}\right.\left.+\delta \lambda_{max}\left(Q\right)\sum_{i=-h}^{-1}\sum_{j=i}^{-1}\zeta^{-j-1}\right\}\\ =\; & \alpha \left(\lambda_{max}\left(P\right)+\eta_{1}\lambda_{max}\left(W\right)\right)+\delta \eta_{2}\lambda_{max}\left(Q\right), \end{array}$$
$$\begin{array}{cc} E\left\{V(k)\right\}>\; & E\left\{\lambda_{min}\left(P\right)x^{T}\left(k\right)x\left(k\right)+\lambda_{min}\left(W\right)\sum_{j=k-h}^{k-1}\zeta^{k-j-1}x^{T}\left(j\right)x\left(j\right)\right\}. \end{array}$$From (Equation 30) and (Equation 31),
$$\begin{array}{cc} \; & \zeta^{N}\left[\alpha \left(\lambda_{max}\left(P\right)+\eta_{1}\lambda_{max}\left(W\right)\right)+\delta \eta_{2}\lambda_{max}\left(Q\right)\right]<\beta \left[\lambda_{min}\left(P\right)+\eta_{1}\lambda_{min}\left(W\right)\right]. \end{array}$$Then,
$$\begin{array}{cc} \; & E\left\{\lambda_{min}\left(P\right)x^{T}\left(k\right)x\left(k\right)+\lambda_{min}\left(W\right)\sum_{j=k-h}^{k-1}\zeta^{k-j-1}x^{T}\left(j\right)x\left(j\right)\right\}\\ <\; & E\left\{\zeta^{N}V\left(0\right)\right\}<\zeta^{N}\left[\alpha \left(\lambda_{max}\left(P\right)+\eta_{1}\lambda_{max}\left(W\right)\right)+\delta \eta_{2}\lambda_{max}\left(Q\right)\right]<\beta \left[\lambda_{min}\left(P\right)+\eta_{1}\lambda_{min}\left(W\right)\right]. \end{array}$$It is easy to see that $E\{x^{T}\left(k\right)x\left(k\right)\}<\beta$, k∈N. Thus, the stochastic system (Equation 1) with h(k)=h is FTS. The proof is completed. □

### 3.4. FTS for Stochastic Systems with Uncertain Parameters

In this section, $e_{1}\left(x\left(k\right),k\right)=e_{2}\left(x\left(k-h\left(k\right)\right),k\right)=0$, and the sufficient conditions of FTS and RAS for stochastic uncertain systems (Equation 1) are presented. At the same time, the FTS of the stochastic system (Equation 1) with h(k)=h is considered.

**Theorem** **3.**
*Given constant ζ>1 and positive constants μ, $\lambda_{i}\left(i=1,2,\cdots ,8\right)$, then the stochastic system (Equation 1) with $e_{1}\left(x\left(k\right),k\right)=e_{2}\left(x\left(k-h\left(k\right)\right),k\right)=0$ is FTS subject to {α,β,N} if there exist matrices $A={\left[A_{1}^{T}\;A_{2}^{T}\;\cdots \;A_{5}^{T}\right]}^{T}$, $B={\left[B_{1}^{T}\;B_{2}^{T}\;\cdots \;B_{5}^{T}\right]}^{T}$ and $C={\left[C_{1}^{T}\;C_{2}^{T}\;\cdots \;C_{5}^{T}\right]}^{T}$ as well as positive definite matrices $P,W_{1},W_{2},Q_{1},Q_{2}\in R^{n\times n}$ satisfying the following conditions*

(36)
$$\left[\begin{array}{cccc} {\left[{\tilde{\Theta }}_{ij}\right]}_{i,j=1,2,\cdots ,5} & \varepsilon_{1}A & \varepsilon_{1}B & \varepsilon_{2}C\\ {}* & -\varepsilon_{1}Q_{1} & 0 & 0\\ {}* & * & -\varepsilon_{1}Q_{1} & 0\\ {}* & * & * & -\varepsilon_{2}Q_{2} \end{array}\right]<0,$$


λ1I<P<λ2I,λ3I<W1<λ4I,λ5I<W2<λ6I,Q1<λ7I,Q2<λ8I,


ζN[α(λ2+η1λ4+η2λ6)+δ(η3λ7+η4λ8)]−β[λ1+η1λ3+η2λ5]<0,

*where*

$$\begin{array}{cc} {\tilde{\Theta }}_{11}= & \;M^{T}PM-\zeta P+W_{1}+W_{2}+{\left(M-I\right)}^{T}Q_{12}\left(M-I\right)+\mu F^{T}F\\ \; & +C_{1}+C_{1}^{T}+D^{T}PD+D^{T}Q_{12}D,\\ {\tilde{\Theta }}_{12}= & \;M^{T}PM_{d}+{\left(M-I\right)}^{T}Q_{12}M_{d}+\mu F^{T}F_{d}+A_{1}-B_{1}+C_{2}^{T},\\ {\tilde{\Theta }}_{13}= & \;B_{1}-C_{1}+C_{3}^{T},\;{\tilde{\Theta }}_{14}=-A_{1}+C_{4}^{T},\\ {\tilde{\Theta }}_{15}= & \;M^{T}PG+{\left(M-I\right)}^{T}Q_{12}G+C_{5}^{T},\\ {\tilde{\Theta }}_{22}= & \;M_{d}^{T}PM_{d}+M_{d}^{T}Q_{12}M_{d}+\mu F_{d}^{T}F_{d}+A_{2}+A_{2}^{T}-B_{2}-B_{2}^{T},\\ {\tilde{\Theta }}_{23}= & \;A_{3}^{T}+B_{2}-B_{3}^{T}-C_{2},\;{\tilde{\Theta }}_{24}=-A_{2}+A_{4}^{T}-B_{4}^{T},\\ {\tilde{\Theta }}_{25}= & \;M_{d}^{T}PG+M_{d}^{T}Q_{12}G+A_{5}^{T}-B_{5}^{T},\\ {\tilde{\Theta }}_{33}= & \;-\zeta^{h_{m}}W_{2}+B_{3}+B_{3}^{T}-C_{3}-C_{3}^{T},\\ {\tilde{\Theta }}_{34}= & \;-A_{3}+B_{4}^{T}-C_{4}^{T},\;{\tilde{\Theta }}_{35}=B_{5}^{T}-C_{5}^{T},\\ {\tilde{\Theta }}_{44}= & \;-\zeta^{h_{M}}W_{1}-A_{4}-A_{4}^{T},\;{\tilde{\Theta }}_{45}=-A_{5}^{T},\;{\tilde{\Theta }}_{55}=G^{T}PG+G^{T}Q_{12}G-\mu I, \end{array}$$

*and the constants $\varepsilon_{1}$, $\varepsilon_{2}$, $\eta_{1}$, $\eta_{2}$, $\eta_{3}$ and $\eta_{4}$ are introduced in (Equation 7) and (Equation 8).*


**Proof.** Let us select the LKF (Equation 9). Next, along the trajectory of system (Equation 1) with $e_{1}\left(x\left(k\right),k\right)=e_{2}\left(x\left(k-h\left(k\right)\right),k\right)=0$, the forward difference of E{ΔV(k)} is obtained as follows:
(37)EΔV(k)≤E(ζ−1)V(k)+xT(k)[MTPM+DTPD+DTQ12D−ζP+(M−I)TQ12(M−I)+W1+W2]x(k)+2xT(k)(MTPMd+(M−I)TQ12Md)x(k−h(k))+2xT(k)(MTPG+(M−I)TQ12G)z(k)+xT(k−h(k))(MdTPMd+MdTQ12Ad)x(k−h(k))+2xT(k−h(k))(MdTPG+MdTQ12G)z(k)+zT(k)(GTPG+GTQ12G)z(k)−ζhMxT(k−hM)W1x(k−hM)−ζhmxT(k−hm)W2x(k−hm)=+ξ˜T(k)(Π˜+ε1AQ1−1AT+ε1BQ1−1BT+ε2CQ2−1CT)ξ˜(k),
where $\tilde{\xi }={\left[\begin{array}{ccccc} x^{T}\left(k\right) & x^{T}\left(k-h\left(k\right)\right) & x^{T}\left(k-h_{m}\right) & x^{T}\left(k-h_{M}\right) & z^{T}\left(k\right) \end{array}\right]}^{T}$,
$$\begin{array}{cc} \tilde{\Pi }=\; & \left[\begin{array}{ccccc} C_{1}+C_{1}^{T} & {\tilde{\Pi }}_{12} & {\tilde{\Pi }}_{13} & -A_{1}+C_{4}^{T} & C_{5}^{T}\\ {}* & {\tilde{\Pi }}_{22} & {\tilde{\Pi }}_{23} & -A_{2}+A_{4}^{T}-B_{4}^{T} & A_{5}^{T}-B_{5}^{T}\\ {}* & * & {\tilde{\Pi }}_{33} & -A_{3}+B_{4}^{T}-C_{4}^{T} & B_{5}^{T}-C_{5}^{T}\\ {}* & * & * & -A_{4}-A_{4}^{T} & -A_{5}^{T}\\ {}* & * & * & * & 0 \end{array}\right], \end{array}$$
$$\begin{array}{cc} {\tilde{\Pi }}_{12} & =A_{1}-B_{1}+C_{2}^{T},\;{\tilde{\Pi }}_{13}=B_{1}-C_{1}+C_{3}^{T},\;{\tilde{\Pi }}_{22}=A_{2}+A_{2}^{T}-B_{2}-B_{2}^{T},\\ \;{\tilde{\Pi }}_{23} & =A_{3}^{T}+B_{2}-B_{3}^{T}-C_{2},\;{\tilde{\Pi }}_{33}=B_{3}+B_{3}^{T}-C_{3}-C_{3}^{T}. \end{array}$$By inequality (Equation 37) and the following condition
(38)$$\begin{array}{cc} \; & \mu x^{T}\left(k\right)H^{T}Hx\left(k\right)+2\mu x^{T}\left(k\right)F^{T}F_{d}x\left(k-h\left(k\right)\right)\\ + & \mu x^{T}\left(k-h\left(k\right)\right)F_{d}^{T}F_{d}x\left(k-h\left(k\right)\right)-\mu z^{T}\left(k\right)z\left(k\right)\geq 0, \end{array}$$
it is inferred that
(39)EΔV(k)≤E(ζ−1)V(k)+ξ˜T(k)(([Θ˜ij]i,j=1,2,⋯,5)+ε1AQ1−1AT=+ε1BQ1−1BT+ε2CQ2−1CT)ξ˜(k).Using the Schur complement property and condition (Equation 36), inequality (Equation 40) holds
(40)$$\begin{array}{c} \left({\left[{\tilde{\Theta }}_{ij}\right]}_{i,j=1,2,\cdots ,5}\right)+\varepsilon_{1}AQ_{1}^{-1}A^{T}+\varepsilon_{1}BQ_{1}^{-1}B^{T}+\varepsilon_{2}CQ_{2}^{-1}C^{T}<0. \end{array}$$From (Equation 39), $E\left\{V\left(k\right)\right\}<\zeta^{k}E\left\{V\left(0\right)\right\}$, k∈N holds. □

For Theorem 3, if ζ=1, the following RAS conditions for the uncertain stochastic system (Equation 1) with TVD are obtained. 

**Corollary** **5.**
*Given constant μ, the stochastic system (Equation 1) with $e_{1}\left(x\left(k\right),k\right)=e_{2}\left(x\left(k-h\left(k\right)\right),k\right)=0$ is RAS if there exist matrices $A={\left[A_{1}^{T}\;A_{2}^{T}\;\cdots \;A_{5}^{T}\right]}^{T}$, $B={\left[B_{1}^{T}\;B_{2}^{T}\;\cdots \;B_{5}^{T}\right]}^{T}$ and $C={\left[C_{1}^{T}\;C_{2}^{T}\;\cdots \;C_{5}^{T}\right]}^{T}$ as well as positive definite matrices $P,W_{1},W_{2},Q_{1},Q_{2}\in R^{n\times n}$, such that all of the following conditions hold:*

$$\left[\begin{array}{cccc} {\left[{\underset{∼}{\Theta }}_{ij}\right]}_{i,j=1,2,\cdots ,5} & (h_{M}-h_{m})A & (h_{M}-h_{m})B & h_{m}C\\ {}* & -\left(h_{M}-h_{m}\right)Q_{1} & 0 & 0\\ {}* & * & -\left(h_{M}-h_{m}\right)Q_{1} & 0\\ {}* & * & * & -h_{m}Q_{2} \end{array}\right]<0,$$

*where*

$$\begin{array}{cc} {\underset{∼}{\Theta }}_{11}= & \;M^{T}PM-P+W_{1}+W_{2}+{\left(M-I\right)}^{T}Q_{12}\left(M-I\right)+\mu F^{T}F\\ \; & +C_{1}+C_{1}^{T}+D^{T}PD+D^{T}Q_{12}D,\\ {\underset{∼}{\Theta }}_{12}= & \;M^{T}PM_{d}+{\left(M-I\right)}^{T}Q_{12}M_{d}+\mu F^{T}F_{d}+A_{1}-B_{1}+C_{2}^{T},\\ {\underset{∼}{\Theta }}_{13}= & \;B_{1}-C_{1}+C_{3}^{T},\;{\underset{∼}{\Theta }}_{14}=-A_{1}+C_{4}^{T},\\ {\underset{∼}{\Theta }}_{15}= & \;M^{T}PG+{\left(M-I\right)}^{T}Q_{12}G+C_{5}^{T},\\ {\underset{∼}{\Theta }}_{22}= & \;M_{d}^{T}PM_{d}+M_{d}^{T}Q_{12}M_{d}+\mu F_{d}^{T}F_{d}+A_{2}+A_{2}^{T}-B_{2}-B_{2}^{T},\\ {\underset{∼}{\Theta }}_{23}= & \;A_{3}^{T}+B_{2}-B_{3}^{T}-C_{2},\;{\underset{∼}{\Theta }}_{24}=-A_{2}+A_{4}^{T}-B_{4}^{T},\\ {\underset{∼}{\Theta }}_{25}= & \;M_{d}^{T}PG+M_{d}^{T}Q_{12}G+A_{5}^{T}-B_{5}^{T},\;{\underset{∼}{\Theta }}_{33}=-W_{2}+B_{3}+B_{3}^{T}-C_{3}-C_{3}^{T},\\ {\underset{∼}{\Theta }}_{34}= & \;-A_{3}+B_{4}^{T}-C_{4}^{T},\;{\underset{∼}{\Theta }}_{35}=B_{5}^{T}-C_{5}^{T},\;{\underset{∼}{\Theta }}_{44}=-W_{1}-A_{4}-A_{4}^{T},\\ {\underset{∼}{\Theta }}_{45}= & \;-A_{5}^{T},\;{\underset{∼}{\Theta }}_{55}=G^{T}PG+G^{T}Q_{12}G-\mu I. \end{array}$$



If h(k)=h, the following corollary is drawn.

**Corollary** **6.**
*Given constant ζ>1 and positive constants μ, $\lambda_{i}\left(i=1,2,\cdots ,5\right)$, then the stochastic system (Equation 1) with $e_{1}\left(x\left(k\right),k\right)=e_{2}\left(x\left(k-h\left(k\right)\right),k\right)=0$ and h(k)=h is FTS subject to {α,β,N} if there exist matrices $A={\left[A_{1}^{T}\;A_{2}^{T}\;A_{3}^{T}\right]}^{T}$ as well as positive definite symmetric matrices $P,W,Q\in R^{n\times n}$, such that all of the following conditions hold:*

(41)
$$\left[\begin{array}{cc} {\left[{\underset{\approx }{\Theta }}_{ij}\right]}_{i,j=1,2,3} & \varepsilon A\\ {}* & -\varepsilon Q \end{array}\right]<0,$$


λ1I<P<λ2I,λ3I<W<λ4I,Q<λ5I,


ζN[α(λ2+η1λ4)+δη2λ5]−β(λ1+η1λ3)<0,

*where*

$$\begin{array}{cc} {\underset{\approx }{\Theta }}_{11}= & \;M^{T}PM-\zeta P+W+h{\left(M-I\right)}^{T}Q\left(M-I\right)+\mu F^{T}F\\ \; & +A_{1}+A_{1}^{T}+D^{T}PD+hD^{T}QD,\\ {\underset{\approx }{\Theta }}_{12}= & \;M^{T}PM_{d}+h{\left(M-I\right)}^{T}QM_{d}+\mu F^{T}F_{d}-A_{1}+A_{2}^{T},\\ {\underset{\approx }{\Theta }}_{13}= & \;M^{T}PG+h{\left(M-I\right)}^{T}QG+A_{3}^{T},\\ {\underset{\approx }{\Theta }}_{22}= & \;M_{d}^{T}PM_{d}-\zeta^{h}W+hM_{d}^{T}QM_{d}+\mu F_{d}^{T}F_{d}-A_{2}-A_{2}^{T},\\ {\underset{\approx }{\Theta }}_{23}= & \;M_{d}^{T}PG+hM_{d}^{T}QG-A_{3}^{T},\;{\underset{\approx }{\Theta }}_{33}=G^{T}PG+hG^{T}QG-\mu I, \end{array}$$

*and the constants ε, $\eta_{1}$ and $\eta_{2}$ are introduced in (Equation 28).*


**Proof.** Consider LKF (Equation 32). Along the trajectory of system (Equation 1), the forward difference of E{ΔV(x(k))} is obtained as follows
EΔV(k)≤E(ζ−1)V(k)+xT(k)[MTPM+DTPD+hDTQD−ζP+h(M−I)TQ(M−I)+W]x(k)+2xT(k)(MTPMd+h(M−I)TQMd)x(k−h)+2xT(k)(MTPG+h(M−I)TQG)z(k)+xT(k−h)(MdTPMd+hMdTQMd)x(k−h)+2xT(k−h)(MdTPG+hMdTQG)z(k)+zT(k)(GTPG+GTQG)z(k)−∑j=k−hk−1ζk−jyT(j)Qy(j)=−ζhxT(k−h)Wx(k−h).From Lemma 1 and the condition (Equation 38), we have
$$\begin{array}{cc} E\left\{\Delta V(k)\right\}\leq \; & E\left\{\left(\zeta -1\right)V\left(k\right)+{\underset{\approx }{\xi }}^{T}\left(k\right)\left(\left({\left[{\underset{\approx }{\Theta }}_{ij}\right]}_{i,j=1,2,3}\right)+\varepsilon AQ_{1}^{-1}A^{T}\right)\underset{\approx }{\xi }\left(k\right)\right\}, \end{array}$$
where $\underset{\approx }{\xi }={\left[\begin{array}{ccc} x^{T}\left(k\right) & x^{T}\left(k-h\right) & z^{T}\left(k\right) \end{array}\right]}^{T}$.By the Schur complement and (Equation 41), the following condition is satisfied
$$\begin{array}{c} \left({\left[{\underset{\approx }{\Theta }}_{ij}\right]}_{i,j=1,2,3}\right)+\varepsilon AQ_{1}^{-1}A^{T}<0. \end{array}$$Then, $E\left\{V\left(k\right)\right\}<\zeta^{k}E\left\{V\left(0\right)\right\}$, k∈N holds. □

## 4. Numerical Examples

This section will provide two simulation examples to demonstrate the validity of the proposed methods.

**Example** **1.**
*Given the coefficient matrices of the discrete-time stochastic system (Equation 1)*


$\begin{array}{c} M=\left[\begin{array}{cc} 0.01 & 0.30\\ 0.20 & 0.00 \end{array}\right],\;M_{d}=\left[\begin{array}{cc} 0.12 & 0.25\\ 0.25 & 0.15 \end{array}\right],\;H=\left[\begin{array}{cc} 0.01 & 0.00\\ 0.00 & 0.01 \end{array}\right],\;H_{d}=\left[\begin{array}{cc} 0.01 & 0.00\\ 0.00 & 0.01 \end{array}\right],\;\\ F=\left[\begin{array}{cc} 0.01 & 0.00\\ 0.00 & 0.01 \end{array}\right],\;F_{d}=\left[\begin{array}{cc} 0.01 & 0.00\\ 0.00 & 0.01 \end{array}\right],\;D=\left[\begin{array}{cc} 0.05 & 0.06\\ 0.05 & 0.01 \end{array}\right],\;G=\left[\begin{array}{cc} 0.01 & 0.00\\ 0.00 & 0.01 \end{array}\right],\;\\ h_{m}=2,\;h_{M}=5,\;\delta =0.2. \end{array}$



The LMIs in Theorem 1 have a feasible solution, the corresponding parameters: μ=0.68, ϵ=6, $\epsilon_{d}=0.83$, $\lambda_{1}=0.4123$, $\lambda_{2}=0.4124$, $\lambda_{3}=0.111$, $\lambda_{4}=0.124$, $\lambda_{5}=0.09$, $\lambda_{6}=0.12$, $\lambda_{7}=0.20$, $\lambda_{8}=0.30$.

Set $\varphi \left(l\right)=\left[\begin{array}{cc} 0.1l+0.2 & 0.1l+0.2 \end{array}\right]$, l∈{−5,−4,⋯,0}, φ(0)=[0.20.2]. $h\left(k\right)=\left\lfloor 3\left|sin(k/15)\right|\right\rfloor$, 2≤h(k)≤5, k∈N, where ⌊·⌋ represents the floor function—that is, adding one after rounding. We observe that the initial value satisfies
$$sup_{l\in \{-5,-4,\cdots ,-1\}}{∥\varphi \left(l\right)∥}^{2}\leq \alpha =3,$$
$$sup_{l\in \{-5,-4,\cdots ,-1\}}{∥\left(\varphi \left(l+1\right)-\varphi \left(l\right)\right)∥}^{2}\leq \delta =0.2.$$

The response of x(k) is plotted in Figure 1, and the evolution of $E\{∥x\left(k\right){∥}^{2}\}$ is plotted in Figure 2. It is not difficult to see that $E\{∥x\left(k\right){∥}^{2}\}\leq 72.3245,\;k\in N$. This implies that system (Equation 1) is FTS subject to (3, 72.3245, 5).

In Table 1, we analyze the uncertain DSNS (Equation 1) with TVD. By Theorem 1, the minimum upper bound of β is obtained for α=3 and N∈(10,20,40).

**Example** **2.**
*Consider the uncertain DSNS (Equation 1)*


$$\begin{array}{l} M=\left[\begin{array}{cc} 0.80 & 0.20\\ 0.07 & 0.10 \end{array}\right],\;M_{d}=\left[\begin{array}{cc} -0.01 & 0.01\\ -0.02 & -0.01 \end{array}\right],\;H=\left[\begin{array}{cc} 0.01 & -0.05\\ -0.02 & 0.01 \end{array}\right],\;\\ H_{d}=\left[\begin{array}{cc} -0.02 & 0.01\\ 0.02 & -0.03 \end{array}\right],\;F=\left[\begin{array}{cc} 0.14 & -0.05\\ -0.04 & 0.05 \end{array}\right],\;F_{d}=\left[\begin{array}{cc} -0.02 & 0.20\\ 0.10 & -0.05 \end{array}\right],\;\\ D=\left[\begin{array}{cc} 0.01 & -0.07\\ -0.01 & 0.01 \end{array}\right],\;G=\left[\begin{array}{cc} 0.01 & 0.00\\ -0.01 & 0.05 \end{array}\right],\;\\ h_{m}=5,\;h_{M}=20,\;\delta =0.12, \end{array}$$

*and the following parameters μ=0.68, ϵ=6, $\epsilon_{d}=0.84$, $\lambda_{1}=0.4123$, $\lambda_{2}=0.4124$, $\lambda_{3}=0.111$, $\lambda_{4}=0.112$, $\lambda_{5}=0.09$, $\lambda_{6}=0.12$, $\lambda_{7}=0.19$ and $\lambda_{8}=0.31$.*


The response of x(k) is plotted in Figure 3, and the evolution of $E\{∥x\left(k\right){∥}^{2}\}$ is plotted in Figure 4.

Moreover, let $h\left(k\right)=\left\lfloor 16\left|sin(k/15)\right|\right\rfloor$, k∈N, 5≤h(k)≤20,
$$\varphi \left(l\right)=\left[\begin{array}{cc} 0.1l+0.2 & 0.1l+0.2 \end{array}\right],l\in \left\{-20,-19,\cdots ,0\right\},$$
satisfy the following conditions
$$sup_{l\in \{-20,-19,\cdots ,-1\}}{E\{∥\varphi \left(l\right)∥}^{2}\}\leq \alpha =2,$$
$$sup_{l\in \{-20,-19,\cdots ,-1\}}{E\{∥\left(\varphi \left(l+1\right)-\varphi \left(l\right)\right)∥}^{2}\}\leq \delta =0.12.$$

Figure 3 and Figure 4 depict the response of x(k) and the evolution of $E\{∥x\left(k\right){∥}^{2}\}$, respectively, for system (Equation 1). One has $E\{∥x\left(k\right){∥}^{2}\}\leq 41.7307,\;k\in N$. Thus, system (Equation 1) is FTS subject to (2, 41.7307, 5).

The minimum upper bound of the parameter β for system (Equation 2) is shown in Table 2, and the following conclusions are obtained:(a)The upper bound of the parameter β in this paper is smaller than that in the literature [41], which infers that the result in this paper is less conservative than that in [41].(b)The authors in [41] employed a novel approximation for a delayed state, which had a smaller approximation error compared with existing approaches. It is worth noting that the system in [41] did not consider the influence of external interference, uncertainty and nonlinear factors.(c)The authors in [13] adopted a new summation inequality called a discrete Wirtinger-based inequality. Although the results of [13] were better than that of this paper, the method adopted in [13] did not reflect random variables and uncertainty. Our results are more general.

The above two examples show that as the TVD increases, the longer it takes for the state response to reach stability. Therefore, the smaller the TVD is, the less conservative the FTS criterion is. It can also be seen that the stability of the state response is independent of the uncertain parameters and is related to nonlinear disturbances and TVD.

## 5. Conclusions

In this paper, we discussed the problem of FTS for uncertain DSNS with TVD. By constructing a new LKF with a power function $\zeta^{k-j-1}$ and new summation inequalities, sufficient conditions to ensure the FTS and RAS of stochastic system were given. By using LMIs, less conservative stability criteria were established. For an uncertain DSNS with h(k)=h, this paper also provided the FTS criteria of the system. Finally, two simulation examples proved the validity of the method. It is noteworthy that the research method proposed can be applied to other systems, for instance, Markov jump systems, singular systems and discrete autonomous systems.

## Figures and Tables

**Figure 1 entropy-24-00828-f001:**
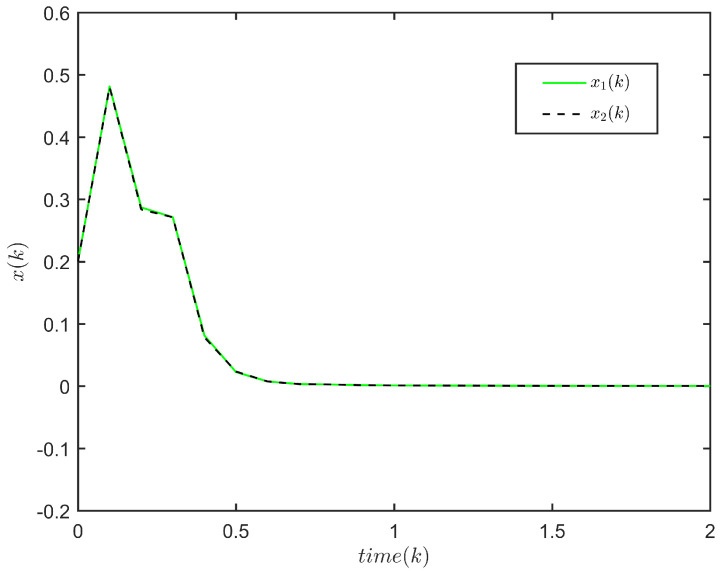
The state response of system (Equation 1) (2≤h(k)≤5).

**Figure 2 entropy-24-00828-f002:**
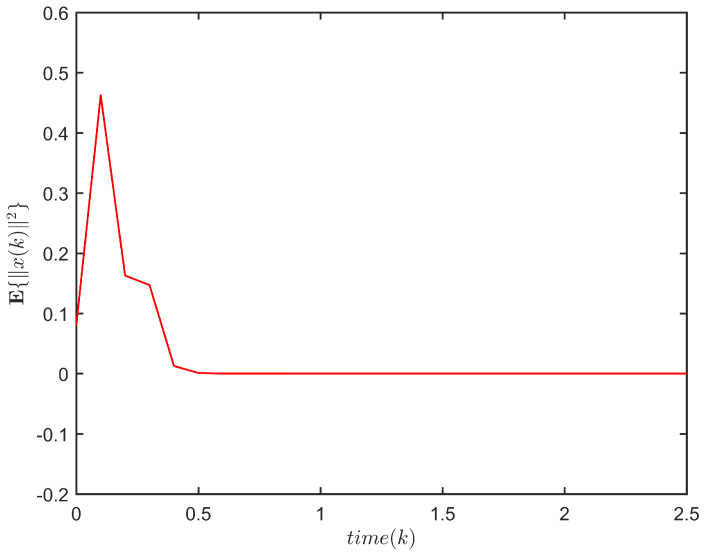
The evolution of $E\{{∥x\left(k\right)∥}^{2}\}$ for (Equation 1).

**Figure 3 entropy-24-00828-f003:**
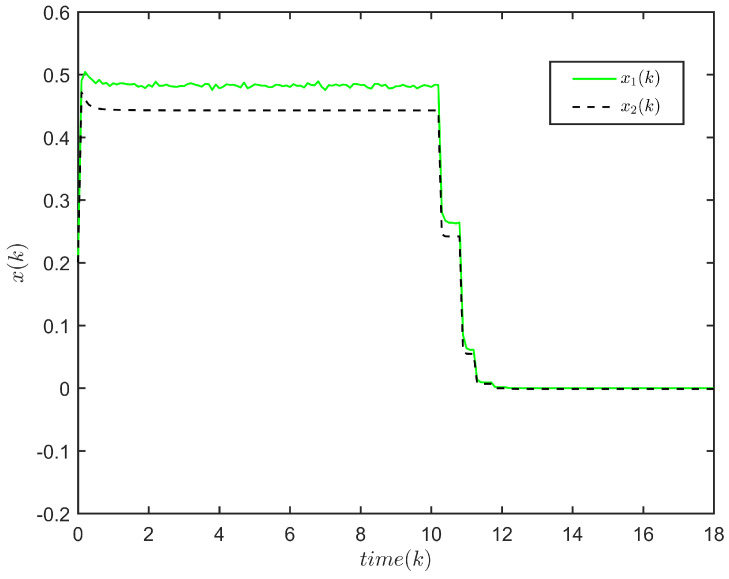
The state response of the system (Equation 1) (5≤h(k)≤20).

**Figure 4 entropy-24-00828-f004:**
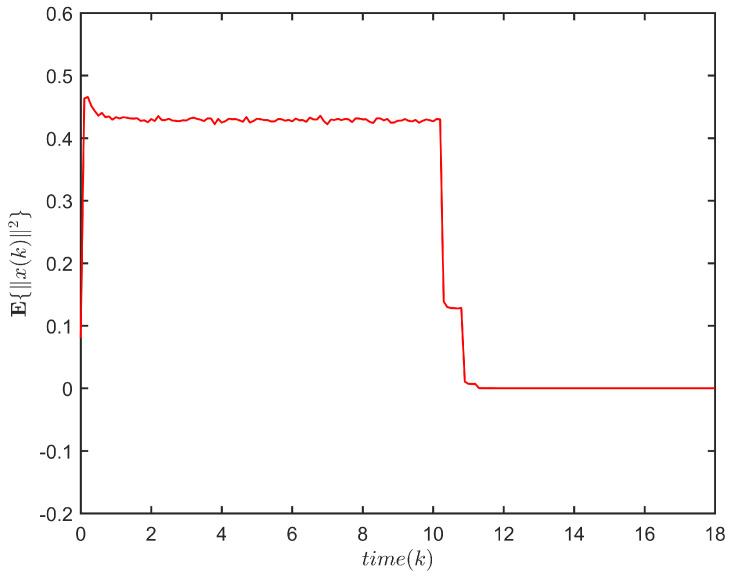
The evolution of $E\{{∥x\left(k\right)∥}^{2}\}$ for system (Equation 1).

**Table 1 entropy-24-00828-t001:** The minimum upper bound of the parameter β for system (Equation 1).

*N*	10	20	40
Theorem 3 [14]	412	$2.11\times 10^{4}$	$4.79\times 10^{7}$
Theorem 1	200.4854	$2.7723\times 10^{3}$	$7.0750\times 10^{5}$

**Table 2 entropy-24-00828-t002:** The minimum upper bound of the parameter β for system (Equation 2).

*N*	5	10	20
Corollary 1 [13]	8	17	19
Theorem 1 [41]	56	125	518
Corollary 1	28.1309	73.9878	424.0102

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
