# Peer review of "Robust Finite-Time Stability for Uncertain Discrete-Time Stochastic Nonlinear Systems with Time-Varying Delay"

_entropy, 2022, doi:10.3390/e24060828_

Round 1
Reviewer 1 Report
The paper concerns the stability of linear discrete-time systems with delay, multiplicative state noise, systems uncertainties and bounded non-linear disturbance.
The problem is solved by means of linear matrix inequalities (LMI).
The technical part is sufficiently correct, accurate and sound.
Overall the novelty and contribution of the paper remains limited. The technical tools use in the paper are quite standard. In the discrete-time case the presence of delay can be avoided by extending the state of the systems, thus the problem can always be transformed in one without delay (see detailed comments below). The LMI-based solution has several drawbacks, for example it does not provide bounds for the delay and it is in any case complex and non constructive. The kind of disturbances considered here are not particularly challenging: all unknown quantities have known bounds, non-linearities are norm-bounded, noises are multiplicative.
Detailed comments
- The class of systems considered here does not have additive noise, thus the effect of noise goes to 0 with x(n), which is restrictive.
- The presence of delay can be modeled in the discrete time case by extending the state with past variables, i.e.
X(n)= [ x(n), x(n-1),...,x(n-hM)]^T.
In this way, the analysis reduces to robust stability of linear systems with uncertainties but no delay. This is why the stability analysis of discrete-time systems has received much less attention (see Remark 1): it is much simpler to solve! - The form (1) is quite specific. Nonlinearities are norm-bounded, uncertainties are bounded by known functions. Isthere a reference to orher papers that have considered this class of systems?
- The use of 'n' as the time index is non standard. Moreover t is confusing because 'n' is the size of the system variables, x(n) in Real^n...
- Definition 1. Does beta depend on alpha? Why is this notion called FTS? Is it not plain stability?
- The use of (x^T)*x throughout the paper does not improve readability. Please consider replacing it with ||x||^2.
- There is something wrong in Lemma 1. What is the thesis? I see only the hypothesis.
- There is no comparison with related methods.
Author Response
We would like to show our gratitude to reviewers for your valuable comments, which help us improve the paper considerably.

Reviewer 2 Report
The submitted manuscript investigates the finite-time stability of stochastic non-linear systems with varying time delays under uncertainties via the Lyapunov-Krasovskii theory and LMIs. Such research extends a bunch of thousands of similar studies. The authors claim that a less conservative estimation of the minimum upper bound on a specific parameter is reached.
The investigated topic of the paper does not attract the reviewer due to the plenty of results published so far; however, it cannot be a reason to reject the manuscript. According to the reviewer’s best knowledge, the presented results have not been published yet. However, their superiority is still questionable (see Comment 1 below).
Comments:
1) While simulation Example 1 suggests a concise comparison to the techniques [14], Example 2 does not provide any proof of the less conservativeness (as promised). Hence, the reviewer would like to read about a comparative numerical study with at least two other methods (if possible).
2) The manuscript should be carefully revised by the authors regarding English and correctness (typos). For instance, see “a uncertain” (line 70) or erroneous subscript “11” in (3) of Lemma 2.
3) Line 149: The reviewer means that the “rounding up” function is usually called the “floor” function.
4) Figures’ legends and axes descriptions should be formatted better. The Italic style, subscripts, etc., should be used as in the text body.
Author Response

(The authors gave the same response as above.)

Reviewer 3 Report
The problem of stability for uncertain discrete-time stochastic nonlinear systems with time-varying delay is too complicated for analytical study. Because of this numerical methodology is developed. The manuscript presents such a methodology for the case of time-varying delay. In the paper a schema for calculation of the corresponding Lyapunov function is presented and it is shown by examples that the formulated criteria for stability of the studied class of stochastic nonlinear systems can work.The study is interesting as it reflects two factors which arise in real situations: the system usually has a stochastic component and the time delay is not strictly constant. I think that the paper is interesting and it can be published.
Author Response
The problem of stability for uncertain discrete-time stochastic nonlinear systems with time-varying delay is too complicated for analytical study. Because of this numerical methodology is developed. The manuscript presents such a methodology for the case of time-varying delay. In the paper a schema for
calculation of the corresponding Lyapunov function is presented and it is shown by examples that the formulated criteria for stability of the studied class of stochastic nonlinear systems can work.The study is interesting as it reflects two factors which arise in real situations: the system usually has a stochastic component and the time delay is not strictly constant. I think that the paper is interesting and it can be published.
Response : Thank you very much for your valuable comments.
Round 2
Reviewer 1 Report
The revised version improved the presentation. Some points were fixed and more explanations were introduced. Although the contribution still suffers from some limitations as reported in the first review, the paper can be accepted for publication.
Author Response
We first would like to show our gratitude to Associate Editor and Referees for your valuable comments, which help us improve the paper considerably.
Thank you very much for your suggestion to the revision of our manuscript. We hope the current version is satisfactory.
Reviewer 2 Report
In the revised manuscript, the authors have corrected most of the issues raised by the reviewer; however, some minor issues remain, as the corrections have not been made carefully.
The comments below refer to those submitted in the reviewer’s previous report:
Ad 2) “A uncertain” appears again in Conclusions (l. 212).
Ad 4) Check the formatting of legends of Figs. 1 and 3 and captions of all independent axes. The legends ignore both Italic style and subscripts. The axes captions do not use Italics for the discrete-time variable.
To sum up, formatting and formal issues still need to be improved.
Author Response
We first would like to show our gratitude to Associate Editor and Referees for your valuable comments, which help us improve the paper considerably.
In the revised manuscript, the authors have corrected most of the issues raised by the reviewer; however, some minor issues remain, as the corrections have not been made carefully.
The comments below refer to those submitted in the reviewer's previous report:
Comment 1: “A uncertain” appears again in Conclusions (l. 212).
Response 1: We are very sorry for the typos. We have corrected it in line 212.
Comment 2: Check the formatting of legends of Figs. 1 and 3 and captions of all independent axes. The legends ignore both Italic style and subscripts. The axes captions do not use Italics for the discrete-time variable.
Response 2: We have carefully checked and corrected them.